# Identification and characterization of a novel enhancer in the HTLV-1 proviral genome

Misaki Matsuo [1,2], Takaharu Ueno [3,12], Kazuaki Monde [4,12], Kenji Sugata[1,12], Benjy Jek Yang Tan [1,2], Akhinur Rahman[1], Paola Miyazato [1,2], Kyosuke Uchiyama [1,2], Saiful Islam [1,2,11], Hiroo Katsuya[1,5], Shinsuke Nakajima[3], Masahito Tokunaga[6], Kisato Nosaka [7,8], Hiroyuki Hata[9], Atae Utsunomiya [6,10], Jun-ichi Fujisawa[3] & Yorifumi Satou [1,2 ✉]

Human T-cell leukemia virus type 1 (HTLV-1) is a retrovirus that causes adult T-cell leuke-mia/lymphoma (ATL), a cancer of infected CD4$^+$ T-cells. There is both sense and antisense transcription from the integrated provirus. Sense transcription tends to be suppressed, but antisense transcription is constitutively active. Various efforts have been made to elucidate the regulatory mechanism of HTLV-1 provirus for several decades; however, it remains unknown how HTLV-1 antisense transcription is maintained. Here, using proviral DNA-capture sequencing, we found a previously unidentified viral enhancer in the middle of the HTLV-1 provirus. The transcription factors, SRF and ELK-1, play a pivotal role in the activity of this enhancer. Aberrant transcription of genes in the proximity of integration sites was observed in freshly isolated ATL cells. This finding resolves certain long-standing questions concerning HTLV-1 persistence and pathogenesis. We anticipate that the DNA-capture-seq approach can be applied to analyze the regulatory mechanisms of other oncogenic viruses integrated into the host cellular genome.

[1] Division of Genomics and Transcriptomics, Joint Research Center for Human Retrovirus Infection, Kumamoto University, Kumamoto 860-8556, Japan. [2] International Research Center for Medical Sciences (IRCMS), Kumamoto University, Kumamoto 860-0811, Japan. [3] Department of Microbiology, Kansai Medical University, Osaka 573-1010, Japan. [4] Department of Microbiology, Faculty of Life Sciences, Kumamoto University, Kumamoto 860-8556, Japan. [5] Division of Hematology, Respiratory Medicine and Oncology, Saga University, Saga 849-8501, Japan. [6] Department of Hematology, Imamura General Hospital, Kagoshima 890-0064, Japan. [7] Department of Hematology, Rheumatology and Infectious Disease, Kumamoto University Hospital, Kumamoto 860-8556, Japan. [8] Cancer Center, Kumamoto University Hospital, Kumamoto 860-8556, Japan. [9] Division of Informative Clinical Sciences, Faculty of Life Sciences, Kumamoto University, Kumamoto 862-0972, Japan. [10] Graduate School of Medical and Dental Sciences, Kagoshima University, Kagoshima 890-8544, Japan. [11] Present address: Viral Recombination Section, HIV Dynamics and Replication Program, National Cancer Institute, Frederick, MD 21702, US. [12] These authors contributed equally: Takaharu Ueno, Kazuaki Monde, Kenji Sugata. ✉email: y-satou@kumamoto-u.ac.jp

Human T-cell leukemia virus type 1 (HTLV-1) is an exogenous retrovirus endemic to some tropical regions[1–3]. HTLV-1 reverse transcribes its viral RNA genome into double-stranded DNA which is then integrated into the host genomic DNA, forming a provirus, which serves as a template for generating new viral particles. A characteristic of HTLV-1 infection is that the virus maintains its copy number during chronic infection not via the production of free viral particles but via clonal expansion and persistence of infected T-cell clones[4,5]. This is the reason the majority of infected individuals remain asymptomatic throughout their lifetime. However, up to 10% of those infected eventually develop adult T-cell leukemia/lymphoma (ATL). ATL can be categorized into two major subtypes: (I) aggressive-type such as acute ATL and lymphoma-type ATL which progress rapidly; and (II) indolent-type ATL such as smoldering ATL and chronic ATL which have a slow disease progression. The pathogenic mechanisms of ATL still remain elusive but is believed to be fueled by the viral genes *tax* and *HBZ* which play key roles in the persistence and expansion of infected cells. Tax, a viral protein encoded in the plus strand of HTLV-1, possesses oncogenic functions such as anti-apoptosis and cell proliferation[6,7] while HBZ, which is encoded in the minus strand and transcribed from the 3′LTR, plays a pivotal role in viral persistence and pathogenesis[4,8]. Studies have shown that there is a constitutively active antisense transcription from the 3′LTR at the population level while sense transcription from the 5′LTR is frequently silenced in ATL cells in vivo[9,10]. This phenomenon suggests that this proviral expression pattern is beneficial for the virus to persist in the host and predisposes infected cells to malignant cellular transformation[5].

We previously reported the presence of an insulator region in the HTLV-1 provirus[11]. While this insulator may explain the contrasting transcriptional pattern between the 5′LTR and 3′LTR of the HTLV-1 provirus, it cannot explain the large difference in their transcriptional activity. In general, exogenous and endogenous retroviruses as mobile DNA elements in the genome can be dangerous to host cells because they may act as genome mutagens and induce genomic instability. Therefore, the host cell has evolved defense mechanisms for transcriptional- and posttranscriptional-silencing of such mobile elements. For example, the KRAB ZnF-Trim28-Setdb1-ZFP809 complex induces transcriptional silencing of murine leukemia virus in embryonic stem cells[12,13]. Constitutive activation of antisense transcription from integrated HTLV-1 provirus may be an exceptional case, indicating the possibility that there may exist a regulatory mechanism that actively maintains transcription from the 3′LTR.

In this study, we screened transcriptional regulatory regions within the HTLV-1 provirus to identify nucleosome-free regions (NFRs) using a highly sensitive micrococcal nuclease sequencing (MNase-seq) approach, followed by our recently developed HTLV-1 DNA-capture-seq protocol[14,15]. The results revealed an internal viral enhancer in the HTLV-1 provirus. It is noteworthy that the region has been intensively analyzed as a coding region of the oncogenic viral gene *tax* without knowing the enhancer function.

## Results

**MNase-seq with HTLV-1 DNA-capture-seq identified a significant nucleosome-free region in the HTLV-1 provirus.** Transcriptional regulatory regions in the genome, such as promoters and enhancers, are generally nucleosome-free because they need to be accessed by transcription factors, epigenetic modifiers, or chromatin remodelers to exert their regulatory functions. We utilized our recently developed HTLV-1 DNA-capture-seq approach, which allows us to increase the detection sensitivity for HTLV-1 sequences up to several thousandfold[14,15]. We first analyzed two HTLV-1-infected T-cell lines, ED and TBX-4B. ED is an ATL cell line derived from an ATL patient, in which sense transcription of the provirus is silenced by DNA methylation and nonsense mutation of the *tax* gene, while the antisense transcription remains active[16,17] (Fig. 1a). On the other hand, TBX-4B, also a T-cell clone derived from an ATL patient; is a non-malignant clone (not derived from ATL cells)[18]. Possibly due to ex vivo cultivation, TBX-4B cells exhibit a higher abundance of HTLV-1 provirus sense transcription compared to antisense strand (Fig. 1a)[19]. To identify previously uncharacterized transcriptional regulatory regions, we screened for NFRs in the HTLV-1 provirus in an unbiased manner by performing MNase-seq analysis, where MNase preferentially digests genomic DNA lacking nucleosomes (Fig. 1b). MNase-seq demonstrated a sharp NFR signal between position 7000–7200 of HTLV-1 in ED cells, close to the insulator region we recently reported[11] (Fig. 1c). Because insulator regions are known to have a regulatory function, they generally possess open (nucleosome-free) chromatin, and thus can be frequently identified using MNase-seq. In ED cells, we observed that the most nucleosome-depleted region falls between the insulator region and the 3′LTR (Fig. 1c). This region is part of exon 3 of the *tax* gene (pX region); however, there have been no previous reports regarding its possible function as a regulatory DNA element. We further asked if the NFR is also observed in in vivo samples in addition to the in vitro cell lines by analyzing peripheral mononuclear cells (PBMCs) freshly isolated from ATL patients and an asymptomatic carrier (AC) (Supplementary Table. 1). Consistent with different DNA methylation status of HTLV-1 provirus between cell lines and in naturally virus-infected cells obtained from PBMCs of infected individuals[17], cells from HTLV-1-infected individuals exhibited higher degrees of nucleosome freeness throughout the provirus in comparison to ED cells. Nevertheless, we discovered that the NFR was also present in the same region as in the HTLV-1 infected cell lines (Fig. 1d), indicating that the NFR is present in vivo in virus-infected cells in individuals as well as in vitro cell lines.

**The nucleosome-free region harbors enhancer-related histone modifications and produces enhancer RNAs.** To investigate the functional role of the NFR, we performed promoter assays using the promoter of the *HBZ* gene[20] (Fig. 2a). Promoter activity was enhanced by the insertion of the NFR either upstream or downstream of the promoter regardless of orientation, indicating that the NFR has an enhancer function (Fig. 2b). We also evaluated the effect of the NFR on the 5′LTR, which is the promoter for sense transcription in the HTLV-1 provirus (Fig. 2c). The promoter activity of the 5′LTR was also enhanced but at a much smaller effect than that observed for the 3′LTR (Fig. 2c). We observed a similar result when we used the U3 region of 5′LTR as a minimal promoter of sense transcription (Supplementary Fig. 1a) which indicated that the NFR exerts its function mainly on the 3′LTR. Since HTLV-1 Tax is known as a transactivator of 5′LTR, we further analyzed the effect of the NFR on the 5′LTR in the presence of Tax and found that Tax increased the enhancer effect just slightly but significantly (Supplementary Fig. 1b). We next analyzed the promoter/enhancer activity with or without the viral CTCF insulator[11]. The presence of the insulator region increased the promoter/enhancer activity (Fig. 2c). We then stimulated the T-cells with TNF-α or Phorbol 12- Myristate 13- Acetate (PMA) but no enhancement of promoter activity (Fig. 2d) is observed, indicating that the NFR enhancer activity is not related to T-cell activation status.

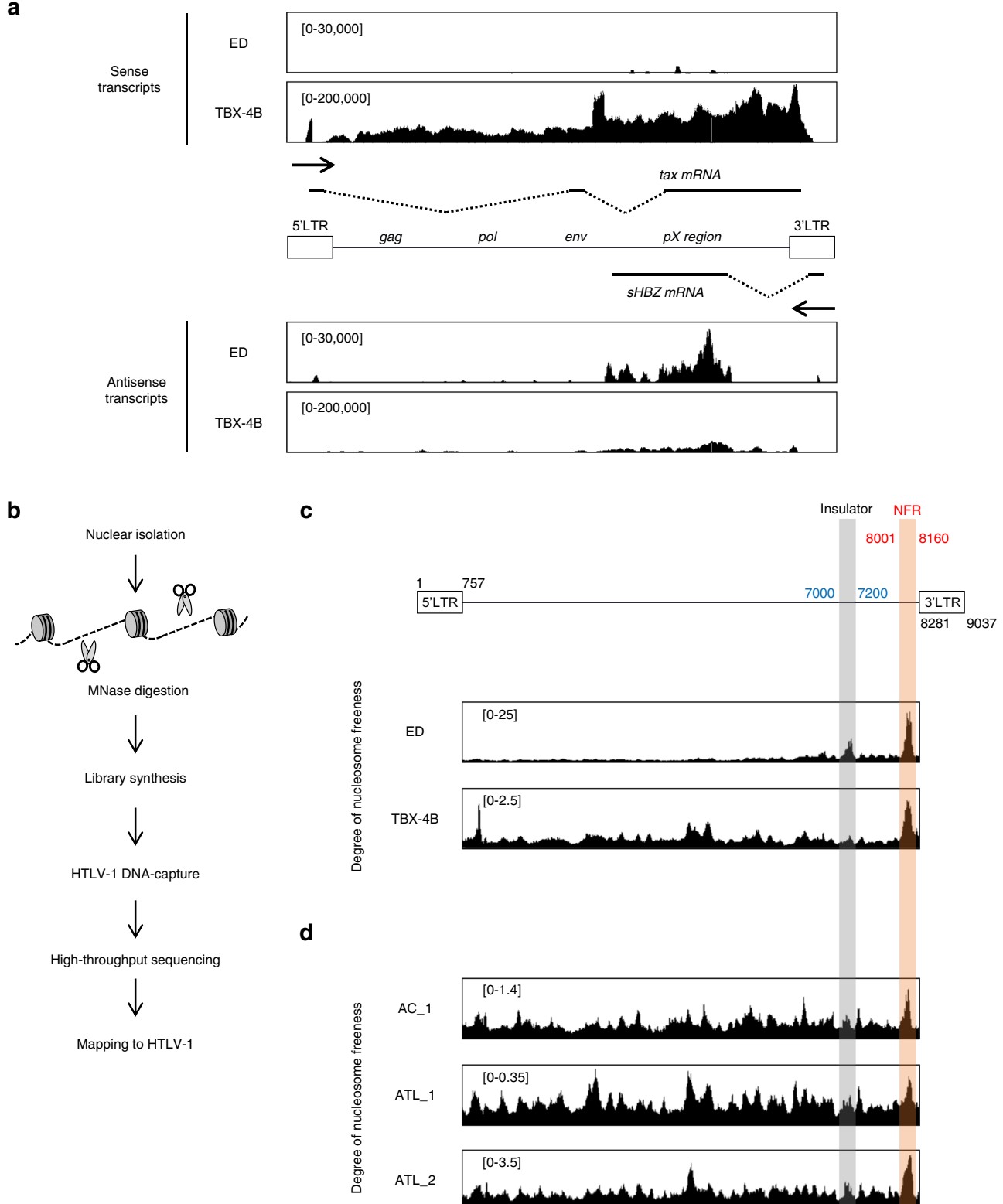

**Fig. 1 Transcriptome pattern and nucleosome positions in the HTLV-1 provirus. a** Proviral transcriptomes are visualized in the sense (top panel) or antisense (bottom panel) orientation in ED cells and TBX-4B cells by Integrative Genomics Viewer (IGV; https://software.broadinstitute.org/software/igv/). The range in square brackets indicates the range of read counts for each transcriptome. **b** Experimental workflow of MNase-seq with HTLV-1 DNA-capture-seq. **c, d** MNase-seq of **c** ED cells and TBX-4B cells and **d** PBMCs of an asymptomatic carrier (AC) and ATL patients. The range in square brackets indicates the degree of nucleosome freeness which is defined as the MNase-seq value normalized to the input DNA-seq value. Orange-shaded region indicates the NFR location and the gray-shaded region indicates the insulator region. The start and end positions of LTRs, insulator, and NFR are indicated as colored numbers (black, blue, and red, respectively).

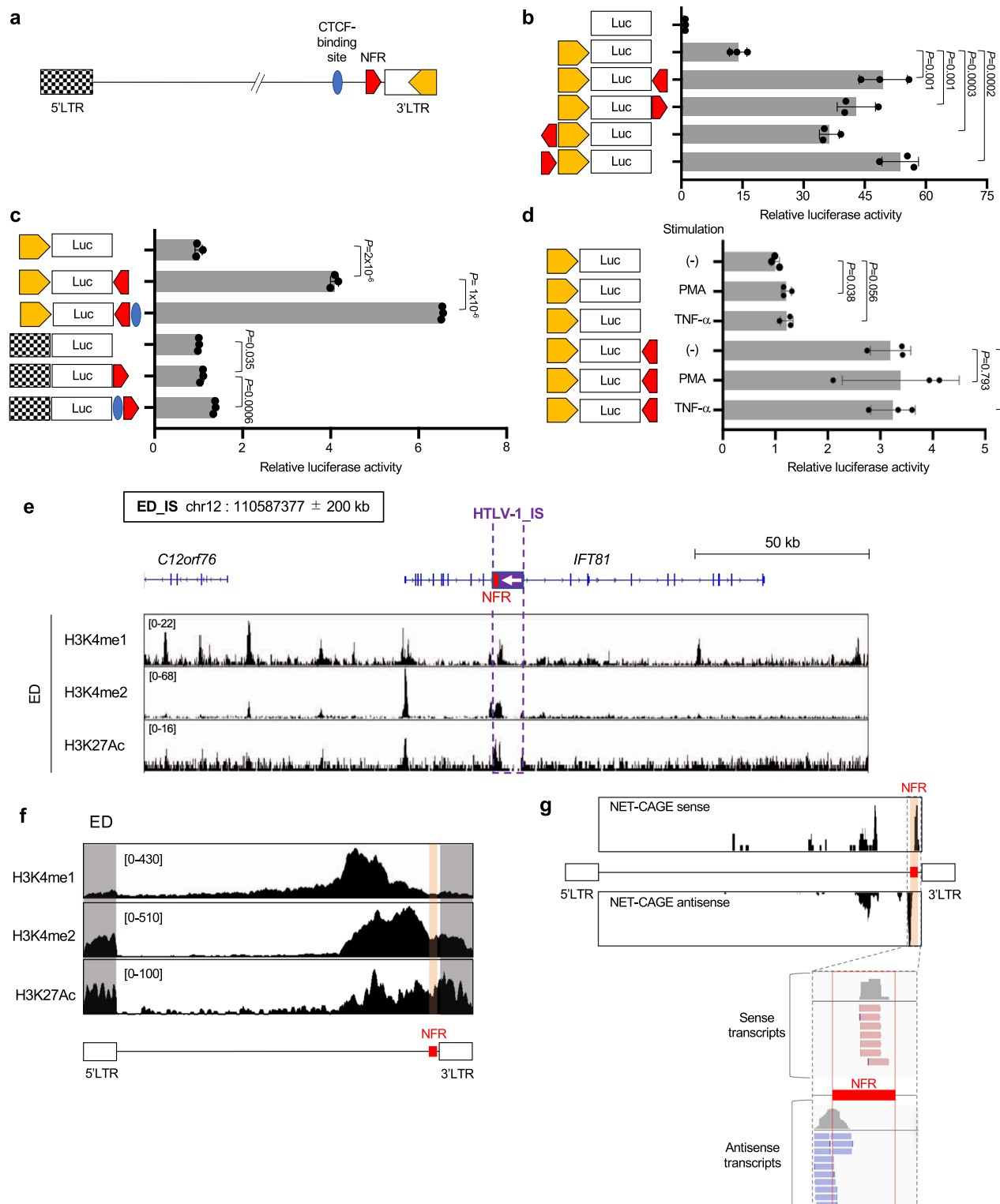

We next analyzed enhancer-related histone modifications within the HTLV-1 proviral region. Chromatin immunoprecipitation sequencing (ChIP-seq) signals of enhancer-related histone modifications[21], including H3K27Ac, H3K4me1, and H3K4me2, showed peaks within a 3-kb distance from the NFR in ED cells (Fig. 2e, f), whereas the peaks were not observed in an infected clone, TBX-4B, without HTLV-1 integration in this locus (Supplementary Fig. 2a). Consistent with the high level of

transcriptional activity from the 5′LTR in TBX-4B cells, in which both the 5′ and 3′LTRs are transcriptionally active (Fig. 1a), there was a wide distribution of enhancer-related histone modifications in this clone, not only in the provirus but also in the host genome nearby the viral integration site (IS) (Supplementary Fig. 2b, c). It has been reported that enhancer regions express enhancer RNAs (eRNAs)—non-coding RNAs with divergent orientation from the center of the enhancer[22]. Thus, we performed a native elongating

**Fig. 2 The nucleosome-free region harbors enhancer-related histone modifications and produces enhancer RNAs. a** Schematic figure of the HTLV-1 provirus structure. The 5′LTR (black plaid), CTCF-binding site (blue)[11], NFR (red), and the HBZ promoter (yellow)[20] are shown. **b**–**d** Transcriptional regulatory function of the NFR was analyzed by luciferase reporter assays in Jurkat cells. **b**, **c** The HBZ promoter[20] and **c** 5′LTR were used as a promoter. **d** The HBZ promoter was used as a promoter and cells were stimulated with either PMA or TNF-α. Luciferase activity was normalized to Renilla activity. Relative luciferase activity is defined as the fold change to pGL4-basic (**b**) or pGL4-basic-HBZ promoter (**c**, **d**). $n = 3$ biologically independent samples, mean ± SD. $P$ values are calculated by a two-sided Student's $t$-test. **e** H3K4me1 (top row), H3K4me2 (middle row), and H3K27Ac (bottom row) ChIP-seq signals near the viral integration site in ED cells. ChIP-seq signals were visualized by IGV. The location (purple square) of provirus and NFR (red square) in ED cells are shown in the upper panel. Provirus direction is represented as a white arrow. The range in square brackets indicates the range of read counts. IS, integration site. **f** H3K4me1 (top row), H3K4me2 (middle row), and H3K27Ac (bottom row) ChIP-seq signals within the provirus in ED cells. ChIP-seq signals were visualized by IGV. Gray-shaded areas and orange-shaded areas indicate the ChIP-seq signals mapped to LTRs and NFR, respectively. The range in square brackets indicates the range of read counts. **g** NET-CAGE of ED cells nuclear lysates in the sense (top row) and antisense (bottom row) orientations, demonstrating eRNAs at the NFR. The bottom panel is an enlarged image of the signals around the NFR. NET-CAGE signals were visualized by IGV.

transcript-cap analysis of gene expression (NET-CAGE) to detect eRNAs[23]. NET-CAGE identifies the sequence of the 5′ region of mRNAs or non-coding RNA adjacent to the cap structure using nascent RNA, which is useful in identifying transcriptional start sites and eRNAs at high resolution. eRNAs from the intragenic HTLV-1 enhancer region were detected in ED cells (Fig. 2g) and TBX-4B cells (Supplementary Fig. 2d). These findings demonstrated that the NFR in the HTLV-1 pX region harbors several fundamental features of an enhancer region.

**The host transcription factors SRF and ELK-1 bind to the intragenic HTLV-1 enhancer**. The NFR region we identified in this study is 160 bp in length. We performed transcription factor binding prediction with the NFR sequence based on the consensus binding motif of various transcription factors and found several candidates (Fig. 3a). We performed a ChIP assay to examine whether these transcription factors localize to the NFR region. Initial evaluation was performed by ChIP-qPCR, and the results were confirmed by ChIP-seq. We found that ELK-1 (Ets like-1 protein) and SRF (serum response factor) clearly bound to the NFR but other transcription factors did not (Fig. 3b and data not shown). Since SRF is also involved in the regulation of the 5′ LTR[24], we also observed the SRF signal in the 5′LTR region in TBX-4B cells, in which HTLV-1 expression in sense orientation is active. Most importantly, the binding of SRF and ELK-1 to the NFR was observed in PBMCs freshly isolated from HTLV-1-infected individuals using highly sensitive ChIP-seq analysis with an HTLV-1 DNA-capture approach[14] (Fig. 3b). These results indicated that this molecular mechanism is actually ongoing in vivo in infected individuals.

Next, we performed electrophoretic mobility shift assays (EMSA) to investigate whether SRF and ELK-1 binding to the NFR depends on DNA sequence. We generated oligonucleotide probes for the NFR using the wild-type (WT) sequence (NFR-wt) and negative control (NC) probes targeting viral regions other than the NFR (Fig. 3c). We observed a band shift when combining the NFR-wt probe and nuclear extract of 293 T cells transfected with SRF and ELK-1 expression vectors (Fig. 3c). Addition of either anti-SRF or anti-ELK-1 antibodies induced a band supershift, demonstrating the involvement of SRF and ELK-1 in the detected band (Fig. 3c). We further generated oligonucleotide probes with mutations in the SRF and/or ELK-1 consensus binding sequence. Mutant_1 (mut_1), mutant_2 (mut_2), and mutant_3 (mut_3) contain mutations in the SRF, ELK-1, or both SRF and ELK-1 binding sites, respectively (Fig. 3d and Supplementary Fig. 3a). To investigate whether the mutations alter transcription factor binding to the NFR, we performed competitive EMSA and found a marked reduction in the binding activity of mutant probes to SRF and ELK-1 compared with that of the wt probe (Fig. 3e). The result showed that there was clear

competition observed with unlabeled wt probes but the competition activity was remarkably decreased when we used mut_1, mut_2, and mut_3 probes. As all mutants showed a marked decrease in the formation of the SRF/ELK-1 ternary complex on the NFR DNA, we used mut_3 for subsequent experiments and found a remarkable reduction in the enhancer activity of the NFR after introducing the mutation (Fig. 3f). These results demonstrated that SRF and ELK-1 binding to the NFR plays an indispensable role in enhancer activity.

**The SRF and ELK-1 play a critical role in the HTLV-1 enhancer function**. Next, we investigated the functional role of SRF/ELK-1 binding to the NFR in the context of the whole viral sequence. As the NFR is located in the coding region of the *tax* gene, we generated mutations in the SRF/ELK-1 binding site without altering the amino acid sequence of the Tax protein. To investigate the possibility that the nucleotide substitutions might alter the *tax* mRNA stability and/or translational efficiency resulting in changes in Tax protein levels, we performed western blotting and confirmed that Tax expression with mut_1, mut_2, or mut_3 sequence was equivalent to that with wt sequence (Supplementary Fig. 3b). We constructed HTLV-1 mutant molecular clones containing the same mutations as mut_3 (Fig. 3d) and then transfected HTLV-1-wt or mut_3 plasmids into 293 T cells. We collected cells one day after transfection and quantified viral gene expression in transfected cells and viral production in the culture supernatant (Fig. 4a). We observed a marginal decrease in p19 production in the supernatant of mut_3 plasmid-transfected cells; however, there was no statistically significant difference (Fig. 4b). Nevertheless, there was a significant reduction in *tax* and *HBZ* mRNA expression (Fig. 4c). Next, we generated T-cell lines infected with HTLV-1-wt or mut_3 by co-culturing the transfected 293 T cells with JET cells—Jurkat T cells stably transfected with a reporter plasmid to monitor Tax expression— as host cells (Fig. 4d). We sorted Tax-expressing cells 3 days after infection and then analyzed provirus sequences, proviral load, and the distribution of HTLV-1 ISs in the sorted cell populations in bulk. We performed DNA sequencing of the whole integrated provirus by DNA-capture-seq and confirmed that the proviral sequences of JET cells infected with HTLV-1-wt and mut_3 were the same as the plasmid sequences used for transfection (Fig. 4e). The proviral load of HTLV-1-mut_3-infected JET cells were lower than that of HTLV-1-wt-infected ones (Fig. 4f), although that may be at least partially induced by the sorting step. Since we sorted infected cells using a reporter fluorescent protein driven by Tax, the cell sorting efficiency in HTLV-1-mut_3 might be lower than that in HTLV-1-wt due to lower *tax* expression (Fig. 4c). We next analyzed whether mutations in the SRF/ELK-1 binding site actually reduced SRF/ELK-1 binding to the NFR in the infected cells in vivo. We performed ChIP-seq analysis for SRF and ELK-1

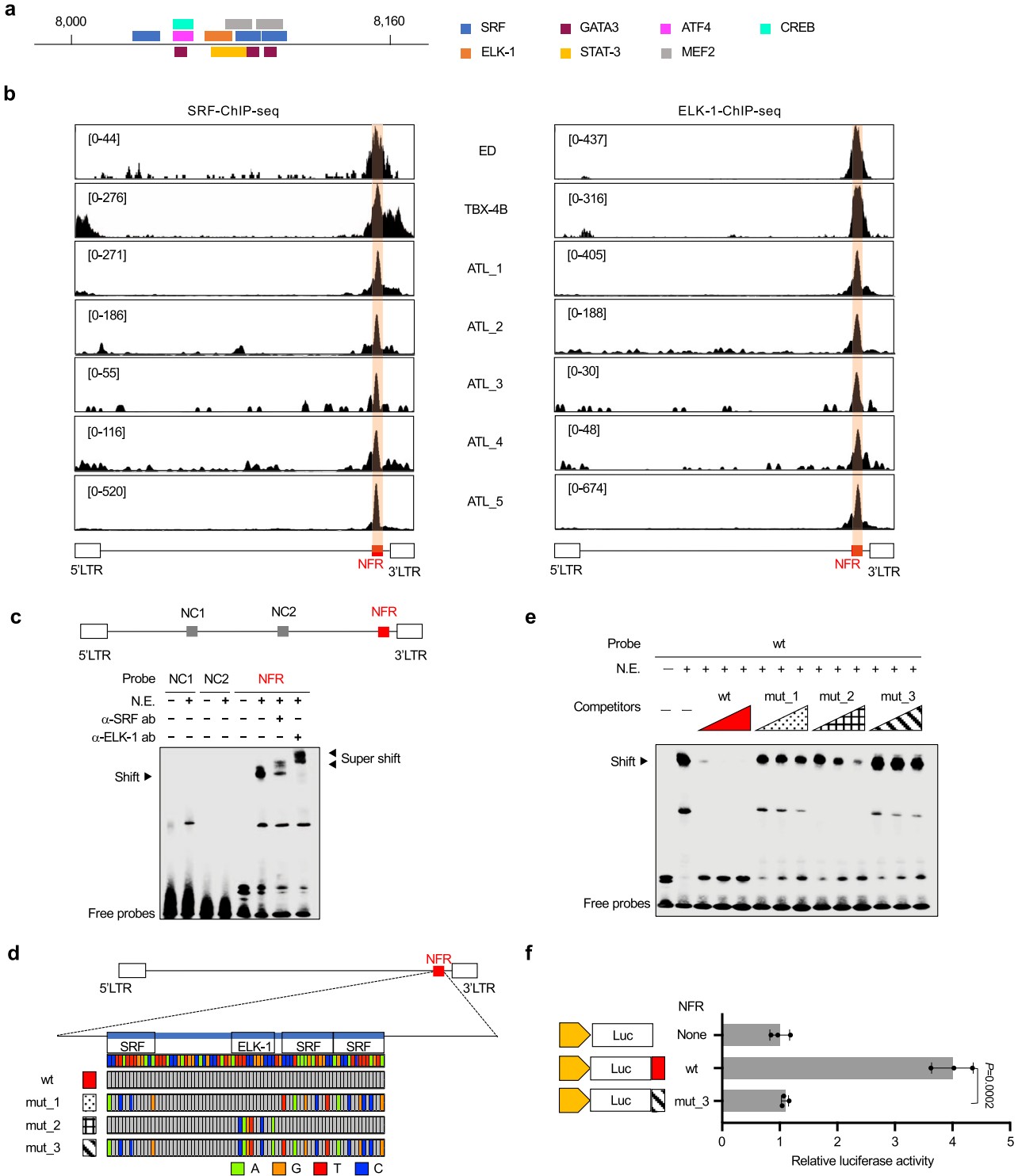

and observed SRF/ELK-1 binding in HTLV-1-wt-infected JET cells but not in mutant virus-infected JET cells (Fig. 4g). Viral IS analysis demonstrated that there were hundreds of different ISs in each condition, i.e., JET cells infected with HTLV-1-wt or mut_3 (Fig. 4h). The distribution of viral IS was not so different between the WT and mutant HTLV-1-infected JET cells in terms of the relationship with the host gene and epigenetic environment (Supplementary Fig. 4a, b and Supplementary Data 1). We then evaluated expression levels of *tax* and *HBZ* in JET cells infected with HTLV-1-wt or mut_3 and found that infected cells with HTLV-1-mut_3 showed a significant reduction

in *tax* and *HBZ* expression (P < 0.05; Fig. 4i). Taking into consideration the similar distribution of ISs between HTLV-1-wt and mut_3-infected cells, the difference in proviral expression can be attributed to the mutation introduced in the NFR of the HTLV-1 provirus and not due to the different distribution of HTLV-1 ISs. HBZ was previously reported to confer an anti-apoptotic phenotype to Jurkat T cells[25]; therefore, we analyzed the susceptibility to apoptosis induced by T-cell activation and found that JET cells infected with HTLV-1-mut_3 were more susceptible to activation-induced T-cell death than those infected with HTLV-1-wt (Fig. 4j). We further analyzed the effect of mutations in the

**Fig. 3 SRF and ELK-1 bind to the NFR in a DNA sequence-dependent manner. a** Prediction of transcription factors that bind to the NFR was performed using TFBIND (http://tfbind.hgc.jp/)[56]. Candidate transcription factor binding sites are shown. 8000 and 8160 are nucleotide positions in HTLV-1 provirus. **b** Figure shows the localization of SRF (left panel) and ELK-1 (right panel) to the NFR in cell lines and PBMCs of ATL patients determined by ChIP-seq with HTLV-1 DNA-capture. ChIP-seq signals were visualized by IGV. The range in square brackets indicates the range of read counts. Orange-shaded region indicates the NFR. **c** Binding ability of SRF and ELK-1 to the NFR oligonucleotides was analyzed by EMSA. Biotinylated DNA probes of 120 bp for the NFR (red) and negative control regions (gray) were incubated with nuclear extract (N.E.) of 293 T cells transfected with SRF and ELK-1 expression vectors. NFR-SRF/ELK-1 complexes and super-shifted complexes, which were detected with the anti-SRF and the anti-ELK-1 antibody, are indicated by arrowheads. The data shown are representative of two independent experiments. **d** Figure shows the mutations introduced to generate three different NFR mutants. The mutated nucleotides are shown in color in comparison to the wt sequence and colored as green, orange, red, and blue for the nucleotides A, G, T, and C respectively. Nucleotide sequences are shown in Supplementary Fig. 3a. **e** EMSA competition analysis with NFR-wt and mutant sequences. Unlabeled competitor NFR-wt or -mut probes were added to the mixture of nuclear lysates and biotin-labeled NFR-wt probe. Competition of unlabeled probes against biotin-labeled NFR-wt probes was evaluated at different doses (x100, x200, and x300). The sequence for each non-labeled competitor is shown in (**d**) and Supplementary Fig. 3a. Red, dots pattern, lattice pattern, and diagonal stripe pattern triangles are indicated as wt, mut_1, mut_2, and mut_3 respectively. For more details, see the method. The data shown were representative of two independent experiments. **f** Transcriptional regulatory function of the NFR-wt (black) or -mut_3 (diagonal stripe pattern) was analyzed using the HBZ promoter (yellow) in Jurkat cells by luciferase assay. Luciferase activity was normalized to Renilla activity. Relative luciferase assay is defined as the fold change compared to a pGL4-basic-HBZ promoter. $n = 3$ biologically independent samples, mean ± SD. $P$ values are calculated by a two-sided Student's $t$-test.

---

NFR on chromatin status and found that the mutations induced a decrease in the chromatin openness of the NFR (Fig. 4k). These findings demonstrate that SRF and ELK-1 binding to the enhancer plays a critical role in the enhancer function.

**The intragenic viral enhancer induces upregulation of host genome transcription near the viral IS.** The presence of an intragenic viral enhancer in the HTLV-1 provirus raises the possibility that it acts as an ectopic enhancer to activate transcription of host cellular genomic DNA, resulting in changes in host gene expression near the viral IS. To investigate the effect of HTLV-1 integration on host gene expression near the ISs, we cloned JET cells infected with HTLV-1-wt or mut_3 by limiting dilution from bulk cell populations (Fig. 4d) and established five clones infected with HTLV-1-wt with one to four proviruses per clone (Fig. 5a). We also established four clones infected with HTLV-1-mut_3 containing one or two proviruses per clone. The characteristics of each individual clone are shown in Supplementary Table 2. We then performed RNA-seq analysis using these clones and found read-through transcripts around the IS of the JET HTLV-1-wt-infected clone (Fig. 5b) but not in the mutant infected clones (Fig. 5c). We further investigated whether the insertion of an ectopic enhancer by HTLV-1 would alter host gene expression. First, we performed principal component analysis (PCA) to investigate global transcriptional differences among JET clones used in the analysis. The result suggested that there was much less global transcriptional difference among each JET clone when compared with TBX-4B and ED cells (Fig. 5d). We analyzed the expression of host genes within the proximity of HTLV-1 provirus, which we defined as genes found within 100 kb up/downstream from the viral IS, and found that the proportion of upregulated genes in JET clones infected with HTLV-1-wt was significantly higher than those in mutant HTLV-1 clones ($P < 0.01$; Fig. 5e). It has been reported that the viral CTCF plays a role in chromatin looping with the host CTCF-binding site and induces changes in host gene transcription[26]. To investigate whether CTCF plays a role in the transactivation of the host gene in cis by HTLV-1 integration, we analyzed CTCF binding to the host genes near ISs. We found a high frequency of CTCF-binding sites in the upregulated host genes, indicating that the transactivation of the host gene by HTLV-1 provirus could be partially mediated by the combination of CTCF and the HTLV-1 enhancer (Fig. 5e). We investigated the possibility that the upregulation of genes near the viral IS was induced via viral gene expression in trans. First, we analyzed the expression level of tax and HBZ genes in each JET clone (Supplementary Fig. 5a). Second, we

analyzed publicly available RNA-seq data regarding Jurkat cells with an inducible *tax* or *HBZ* gene to investigate whether the genes we show in Fig. 5e would be induced by *tax* or *HBZ* expression (Supplementary Fig. 5b)[27]. These data indicated that upregulation of the host genes near viral IS was not via viral gene expression but at least partially mediated by an ectopic presence of the enhancer inserted by HTLV-1. Further experiments are needed to understand the effect of HTLV-1 integration on the host genome.

The HTLV-1 ISs are different among JET clones infected with HTLV-1-wt or mut_3, therefore, we cannot exclude the possibility that different IS may generate the different transcriptome in the clones we analyzed in Fig. 5. To solve this issue, we introduced mutations that abrogate the SRF/ELK-1 binding to the enhancer region (Figs. 3d, e and 4e) in a clone infected with HTLV-1-wt (wt_#5, Supplementary Table 2) by using CRISPR/Cas9 technique. SRF/ELK-1 ChIP-seq peaks in HTLV-1-wt-infected cells were abolished in the CRISPR-mutated cells (C-mut), thereby reducing proviral transcription in both sense and antisense strands (Fig. 6a–c) as well as chromatin openness in the enhancer region (Fig. 6d). We further analyzed another clone with two copies of HTLV-1-wt provirus (wt_#1, Supplementary Table 2). We searched the CRISPR-mutated clones with the mut_3 sequence in the enhancer region of two proviruses by doing the whole proviral sequence by HTLV-1 DNA-capture-seq and identified the clone with mut_3 sequence in the NFR in two proviruses while the other regions were identical to the wt_#1 cells. We performed RNA-seq analysis and found that proviral transcription was remarkably decreased (Fig. 6e–g). Importantly, there were a clear decrease in the host transcriptome and splice junction near the viral integration site in C-mut clone (Fig. 6h, i). These data collectively provided the evidence to support the idea that SRF- and ELK-1-binding to the NFR play an important role in the enhancer function.

**SRF and ELK-1 localization to the enhancer and aberrant host genome transcription near the proviral integration site in fresh PBMCs.** We further investigated the effect of HTLV-1 integration on viral and host genomes by performing mRNA-seq analysis using freshly isolated PBMCs from five ATL cases. All five cases had a high proviral load (Supplementary Table 1) and had a clonally expanded ATL clone (Supplementary Fig. 6). Consistent with the previous reports[9,28], proviral expression in the sense orientation was lower than that in the antisense orientation (Fig. 7a). There was read-through proviral transcription in the sample with HTLV-1 ISs in the host genomic region but not in

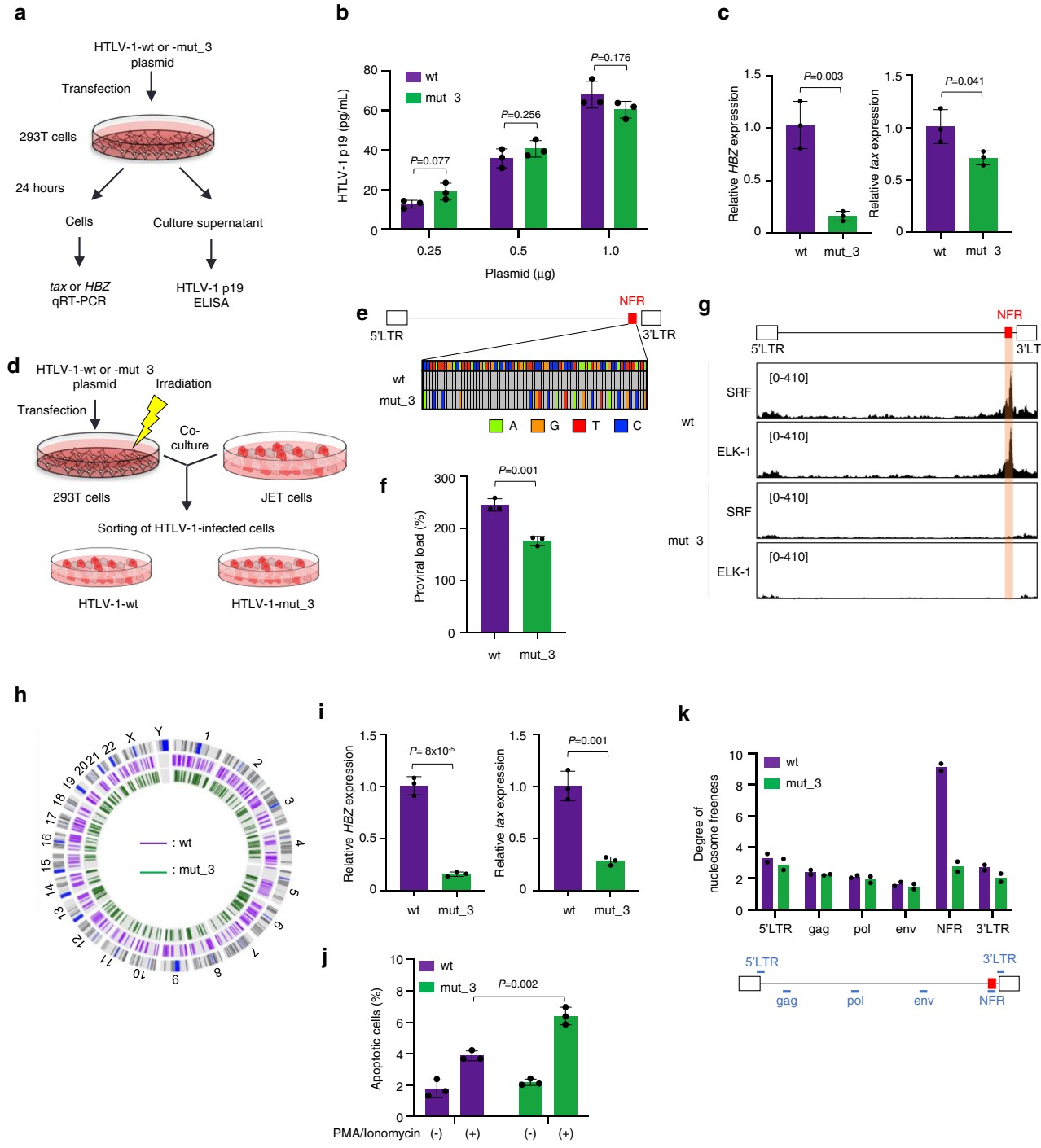

other samples (Fig. 7b, c), as previously reported[29]. Interestingly, an ATL case with a defective provirus lacking the 5′LTR also exhibited read-through transcription from the virus to the flanking host genome (Fig. 7d). More importantly, there were clear peaks of SRF and ELK-1 ChIP-seq signals in the integrated proviruses, indicating SRF and ELK-1 play a role in the transcriptional regulation (Fig. 7b–d). PBMCs contain not only ATL cells but also non-ATL infected T cells, uninfected T cells, and various non-T cells; thus, the mRNA-seq data shown in Fig. 7b–d represents the average expression of PBMC subsets. To observe the effect of HTLV-1 ISs on the host genome with high accuracy at single-cell resolution, we performed single-cell RNA-seq analysis using PBMCs from five ATL cases including the same ATL case as in Fig. 7b, c, and in other three ATL cases containing

defective proviruses. Based on the T-cell receptor (TCR) clonotype and transcriptome data, we performed clustering analysis and found that the ATL clones, which were identified by the T-cell receptor (TCR) clonotype, clustered differently from the other CD4+ T-cell clones (Fig. 7e, f). We then compared the transcriptome near viral IS of CD4+ T cells among five ATL cases. There was remarkable upregulation of the local transcriptome only in the sample with viral integration (Fig. 7g, h and Supplementary Fig. 7a–c, left panels), which is consistent with previous reports showing read-through transcript from defective provirus[28,29]. Furthermore, there was a significant increase in the local transcriptome in the ATL clone but not in non-ATL CD4+ T-cell clones (Fig. 7g, h and Supplementary Fig. 7a–c, right panels). These data were consistent with the idea that the

**Fig. 4 Generation and characterization of the HTLV-1 infectious clones with NFR-wt or mut_3 in the SRF/ELK-1 binding sites. a** Diagram illustrating the experimental workflow for evaluating transient transfection using HTLV-1-wt or mut_3 molecular clones. **b** HTLV-1 p19 was quantified with ELISA using the supernatant of 293 T cells transfected with HTLV-1-wt or mut_3 (0.25, 0.5, 1 µg). Results are shown as the mean ± SEM of three experiments performed in duplicates. $P$ values are calculated by a two-sided Student's $t$-test. **c** qRT-PCR results of HBZ (left) and tax (right) after transient transfection. 18 S rRNA was used as an internal control. $n = 3$ biologically independent samples, mean ± SD. $P$ values are calculated by a two-sided Student's $t$-test. **d** Experimental workflow illustrates the establishment of stable cells infected with HTLV-1-wt or -mut_3. HTLV-1-infected cells were sorted using tdTomato driven by Tax-responsive elements as a marker. **e** The HTLV-1 NFR sequences of JET cells infected with HTLV-1-wt or mut_3 were analyzed by HTLV-1 DNA-capture-seq. The mutated nucleotides (A, G, T, and C) in comparison to the wt sequence are shown in green, orange, red and blue, respectively. **f** Proviral load in JET cells infected with HTLV-1-wt or mut_3 was measured by digital droplet PCR. $n = 3$ biologically independent samples, mean ± SD. $P$ values are calculated by a two-sided Student's $t$-test. **g** SRF and ELK-1 ChIP-seq signals in JET cells infected with HTLV-1-wt (top 2 rows) or -mut_3 (bottom 2 rows) were visualized by IGV. The range in square brackets indicates the range of read counts. The orange-shaded area represents the NFR location.
**h** Distribution of integration sites (ISs) in bulk JET cells infected with HTLV-1-wt (purple line) or -mut_3 (green line) is shown in a circos plot. The outer ring with numbers and letters represents the human chromosomes with the cytogenetic bands shown as gray lines and the centromere shown as a blue line. **i** Representative qRT-PCR results of HBZ (left) and tax (right) levels in JET cells infected with HTLV-1-wt or mut_3. 18 S rRNA was used as an internal control. $n = 3$ biologically independent samples, mean ± SD. $P$ values are calculated by a two-sided Student's $t$-test. **j** Cell apoptosis was detected by Annexin V staining after stimulation with PMA and ionomycin. Bar graphs were generated with Annexin V staining positive cells in the alive cell population gated by FSC/SSC dot plots in supplementary Fig. 8. $n = 3$ biologically independent samples, mean ± SD. $P$ values are calculated by a two-sided Student's $t$-test. **k** MNase assay of JET cells infected with HTLV-1-wt or mut_3. The degree of nucleosome freeness was evaluated by MNase treatment and ddPCR. Values of indicated proviral regions are shown after normalization to the values from non-MNase digestion samples. $n = 2$ biologically independent samples.

intragenic viral enhancer we identified in this study plays a role in the persistent proviral expression and aberrant transcription of the integrated host genome by recruiting SRF and ELK-1.

## Discussion

The HTLV-1 genome is just over 9000 bp in size but by alternative splicing, it encodes several viral genes which play a role to help the virus achieve persistent infection in the host. Additionally, the provirus is transcribed from both the 3′LTR and the 5′LTR[20,30,31]. It has been reported that antisense transcription is frequently expressed in vivo, whereas sense transcription is typically silenced or expressed only intermittently[9,17,32,33]. It has not been understood how HTLV-1 antisense transcription remains selectively active. In the present study, we demonstrated the presence of a previously uncharacterized viral enhancer in the HTLV-1 pX region by exploiting the high efficiency and resolution of the viral DNA-capture-seq approach. The enhancer we identified here is located at the 3′ side of the insulator region in the provirus (Fig. 2a). This enhancer is not a typical retroviral enhancer as retroviral enhancers are generally located in the LTR region[30,34]. Thus, we propose that this internal enhancer region near the 3′LTR may have two distinct functions: first, to drive the frequent antisense transcription from the 3′LTR, and second, to co-operate with the viral insulator to inhibit the spread of heterochromatin from the 5′LTR towards the 3′LTR. The antisense transcript HBZ plays an indispensable role in viral persistence[9,35] and therefore the intragenic viral enhancer would also contribute to viral persistence via HBZ upregulation. Consistent with this notion, the intragenic viral enhancer and insulator are maintained even in defective type proviruses that are observed in 20–30% of ATL cells[15,36,37]. Further experiments are required to understand how these viral regulatory elements, including 5′LTR, viral insulator, enhancer, and 3′LTR, in the small viral genome cooperatively regulate viral and host genome transcriptome.

There are several thousands of different HTLV-1-infected T-cell clones in an infected individual[38]. After long-term clinical latency, a specific clone may undergo malignant transformation, causing the syndrome of ATL. A key question that remains is how a certain clone is selected as an ATL clone from various infected clones. Previous reports demonstrated that the HTLV-1 infected clones harboring proviruses near cancer-related genes are preferentially selected in ATL cells[29,39], indicating that aberrant host genome transcription caused by viral integration may contribute

to the multistep oncogenic process induced by HTLV-1 infection. HTLV-1 contains a CTCF-binding site and therefore viral integration generates an ectopic CTCF-binding site in the host genome[11], which induces deregulation of host gene transcription via chromatin looping as Melamed A et al reported[26]. We demonstrate here that HTLV-1 generates an ectopic enhancer region in addition to the viral CTCF insulator region. These findings indicate that the HTLV-1 enhancer can induce a distinct alteration of the host transcriptome via chromatin looping[26], and thereby upregulates cancer-related genes near ISs which might contribute to the preferential selection of a specific infected cell for clonal expansion during the early phase of leukemogenesis. This mechanism is similar to how endogenous retroviruses might contribute to the development of acute myeloid leukemia[40].

Mobile DNA elements, including endogenous retroviruses or foreign DNA elements introduced by exogenous retroviruses, can be dangerous for the host cell because they disturb cellular genomic homeostasis. Mammalian cells have an evolutionarily acquired host defense system that silences such elements in the genomic DNA. For example, the murine leukemia virus (MLV) is silenced by Trim28—a well-characterized transcriptional co-repressor[41]—and ZFP809 to prevent further viral spread in embryonic stem cells[13]. Although little is known regarding the precise molecular mechanisms behind the silencing of HTLV-1 provirus in the host genome, the HTLV-1 5′LTR is frequently silenced by DNA methylation, histone modifications[42], or only transcribed intermittently[32,33]. This suggests that a host defense mechanism is playing a role in selecting infected clones with silenced HTLV-1 proviral DNA. As a result, there is no detectable viremia in the serum of HTLV-1-infected individuals. However, the virus maintains the ability to reactivate viral transcription when the virus needs to induce de novo infection from an infected host to an uninfected host. We showed here that HTLV-1 recruits the host transcription factors SRF and ELK-1 to an NFR in proviral DNA to sustain chromatin openness and proviral transcription in host cells. This molecular mechanism possibly enables the virus to be latent but at the same time allows it to maintain the ability to reactivate viral expression when infected cells need to induce de novo infection from the infected to the uninfected host.

HTLV-1 has co-existed with humans for the past 20,000–30,000 years[43]. The virus may have evolved this strategy—the presence of an internal insulator and enhancer region in the

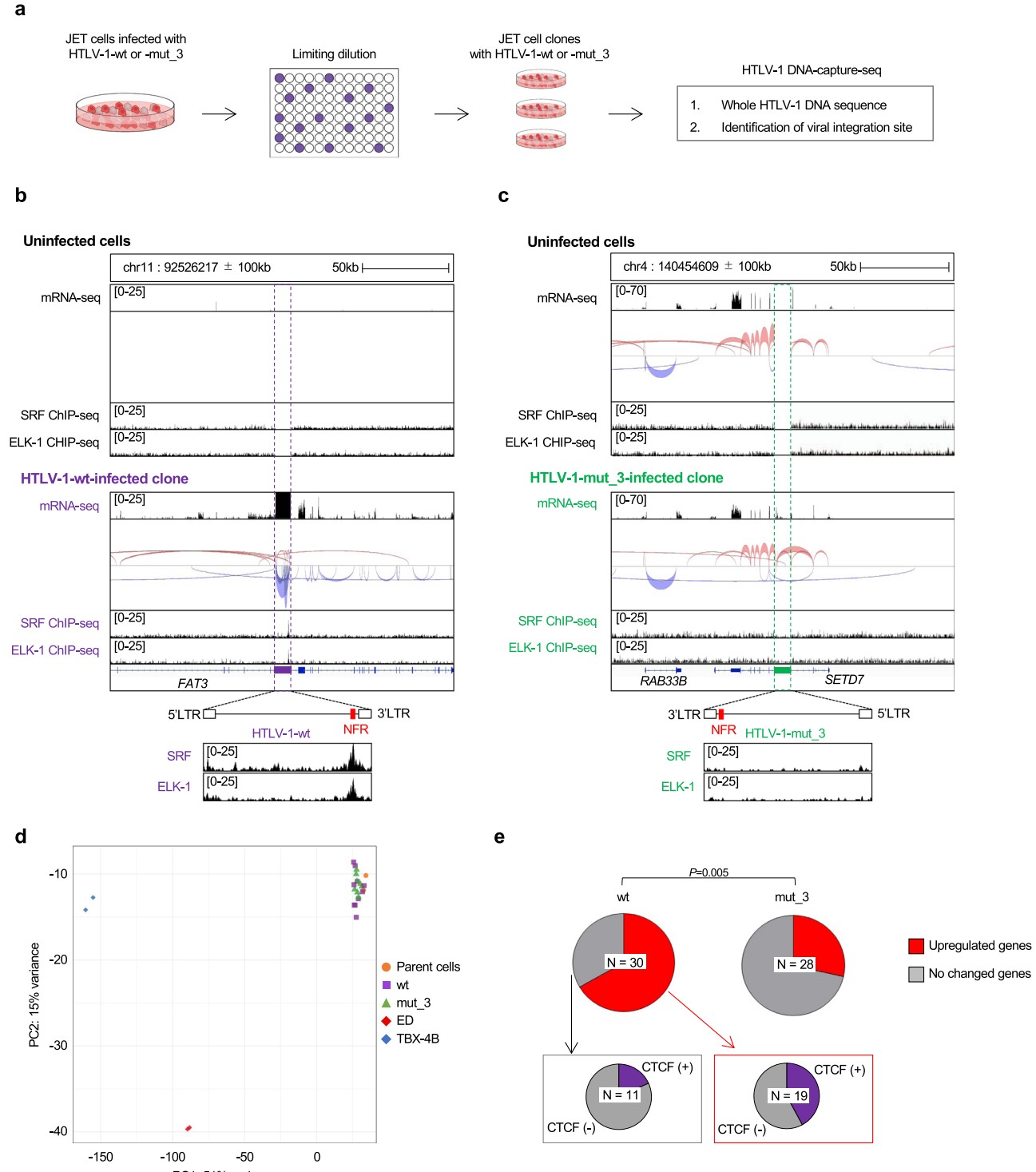

**Fig. 5 Establishment and characterization of JET cell clones infected with HTLV-1-wt and mut_3. a** Experimental workflow to establish clones infected with HTLV-1-wt or mut_3 by limiting dilution. **b**, **c** Local transcriptome and splice junction near the viral integration site in a JET clone infected with **b** HTLV-1-wt (wt_#1) and **c** mut_3 (mut_3_#1) are visualized by IGV. The splice junctions are shown in red for sense transcripts and blue for antisense transcripts. The line thickness indicates the frequency of specific splice patterns detected. Host genes near the IS and direction of HTLV-1 provirus are shown in the lower panel. SRF/ELK-1 ChIP-seq results are also shown for each clone in the bottom row. **d** Differential expression analysis in cell lines (ED and TBX-4B), parent cells (JET), wild type, and mutated clones (see Supplementary Table 2) visualized by principal component analysis (PCA) plot. Gene expression levels of each cell were quantified using mRNA-seq data performed in duplicates. The percentage variance for each PC were shown on the respective PC axis. **e** The fraction of upregulated genes in JET clones infected with HTLV-1-wt (left pie) and mut_3 (right pie). The analysis was performed with genes within 100 kb from viral IS. The presence or absence of CTCF ChIP-seq signals in the upregulated group (bottom left) or "no change" group (bottom right) of JET clones infected with HTLV-1-wt. *P* value was calculated by a two-sided Chi-square test.

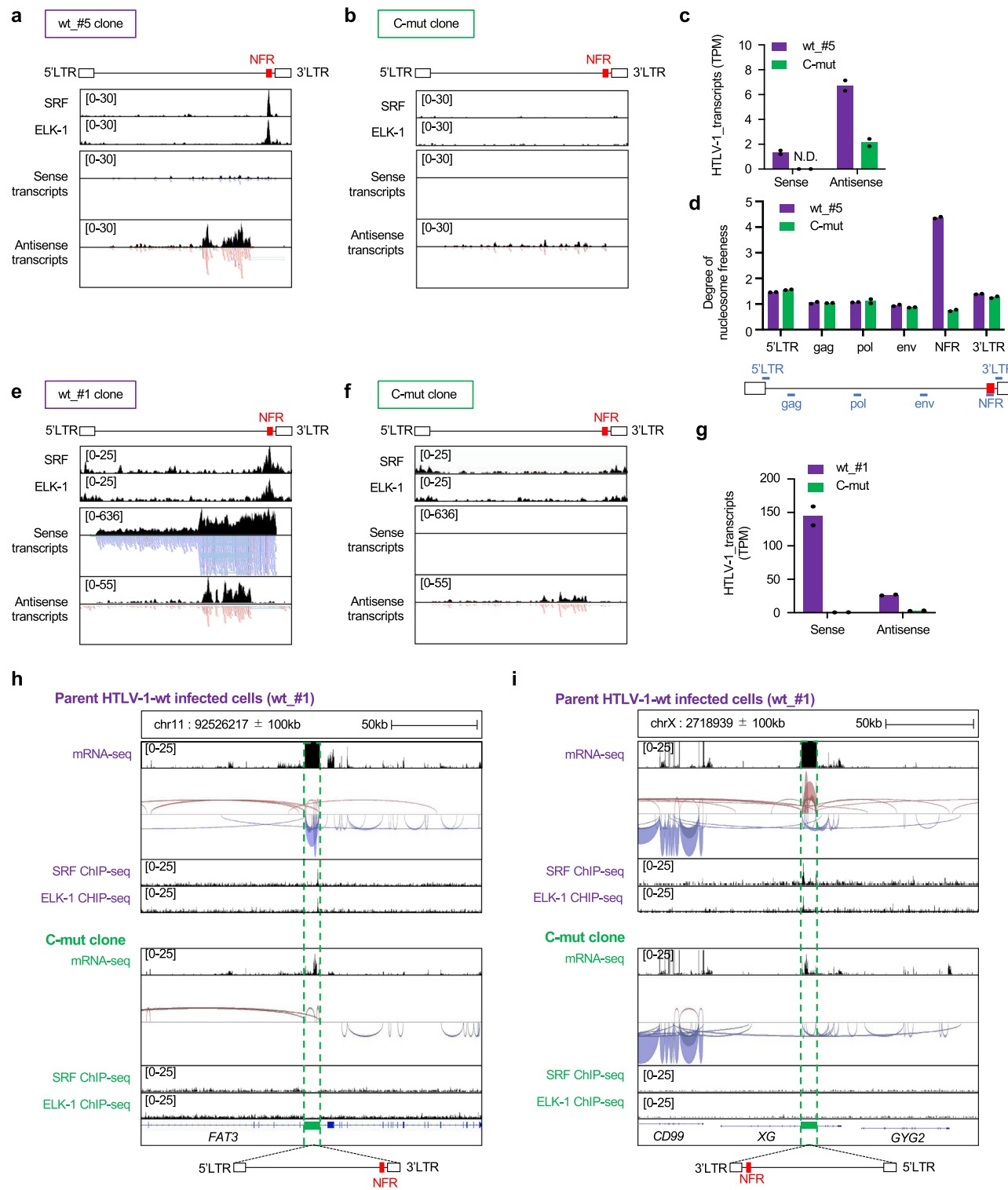

provirus—to achieve persistent infection under pressure from the host system to silence foreign DNA elements as well as from the host immune response. Usage of lentiviral/retroviral vectors for gene therapy or for the generation of induced pluripotent stem (iPS) cells has been under intense research and development[44]. Lentiviral and retroviral vectors integrate into host genomic DNA and form a provirus in the target cells; however, the provirus tends to be silenced by host defense mechanisms as described above. Various efforts have been made to optimize the lentiviral and retroviral vectors to prevent the silencing of the integrated

provirus, such as the introduction of insulator or enhancer elements[44,45]. Retrovirus vector insertion can trigger deregulated cell proliferation, most likely driven by the activity of retrovirus enhancers on cancer-related genes[46]. It is surprising that an exogenous virus HTLV-1 has by itself evolved a similar system, obtaining an insulator, an enhancer, and a chromatin-opening element in the retroviral genome. This experiment of nature may provide insights into how an exogenous retrovirus achieves persistent infection in humans and also how to tackle the silencing of foreign DNA elements to maintain chromatin openness and

**Fig. 6 Establishment and characterization of HTLV-1-mut_3 clones by CRISPR/Cas9. a, b** mRNA-seq and SRF/ELK-1 ChIP-seq results of **a** HTLV-1-wt (wt_#5) and **b** CRISPR-mutated HTLV-1-mut_3 (C-mut) clone. Representative results from two independent experiments are visualized by IGV. The range in square brackets indicates the range of read count. **c** Level of proviral expression of J HTLV-1-wt (wt_#5) or CRISPR-mutated (C-mut) clones. Data were shown as transcripts per million reads (TPM) from two independent mRNA-seq analyses. **d** MNase assay of JET clones infected with HTLV-1-wt (wt_#5) or C-mut. The degree of nucleosome freeness was evaluated by MNase treatment and ddPCR. Values of indicated proviral regions are shown after normalization to values from non-MNase digestion samples. $n = 2$ biologically independent samples. **e, f** mRNA-seq and SRF/ELK-1 ChIP-seq results of JET cells infected with **e** HTLV-1-wt (wt_#1) and CRISPR-mutated **f** HTLV-1 (C-mut). Representative results from two independent experiments are visualized by IGV. The range in square brackets indicates the range of read counts. **g** Level of proviral expression of JET cells infected with HTLV-1-wt (wt_#1) or C-mut clones. Data were shown as transcripts per million reads (TPM) from two independent mRNA-seq analyses. **h, i** Local transcriptome and splice junction near the viral integration site are visualized in a JET cell clone infected with HTLV-1-wt (wt_#1) and C-mut by IGV in each integration site; **h** chr11 and **i** chrX. The splice junctions are shown in red for sense transcripts and blue for antisense transcripts. The line thickness of splice junctions indicates the frequency of specific splice patterns detected. Host genes near the IS and direction of HTLV-1 provirus are shown in the lower panel. SRF/ELK-1 ChIP-seq results are also shown for each clone in the bottom row.

---

transgene transcription without causing the transformation of host cells.

In conclusion, we have analyzed the HTLV-1 provirus integrated into the host genome with high resolution and efficiency using the HTLV-1-DNA-capture sequencing approach and discovered an internal viral enhancer in the HTLV-1 genome. This finding provides clues to help solve several long-lasting questions related to HTLV-1 persistence and pathogenesis. Viral DNA-capture-seq approaches can be applied to studies aiming to understand the transcriptional regulatory mechanism of other oncogenic viruses integrated into the host cellular genomic DNA.

## Methods

**Ethics statement and patient blood samples.** All protocols involving human subjects were reviewed and approved by the Kumamoto University Institutional Review Board (approval number 263). The study was carried out in accordance with the guidelines proposed in the Declaration of Helsinki. Informed written consent was obtained from all subjects in this study. Peripheral blood mononuclear cells (PBMCs) were isolated from whole blood within 24 h of sample collection using Ficoll-Paque (GE Healthcare Life Sciences, Marlborough, MA) according to the manufacturer's instructions. Characteristics of clinical samples were summarized in Supplementary Table 1.

**Cell culture.** JET cells[47] are Jurkat cells expressing tdTomato under the control of five times tandem repeat of Tax-responsive element (TRE). ED[16], 293 T, Jurkat, and JET cells infected with WT or mutant HTLV-1 molecular clones were cultured in RPMI-1640 medium (Thermo Fisher Scientific, Waltham, MA) supplemented with 10% fetal bovine serum (FBS), 100 U/mL penicillin, and 100 μg/mL streptomycin. TBX-4B cells[18] were cultured in RPMI-1640 supplemented with 20% FBS, interleukin-2 (200 U/mL; PeproTech, Cranbury, NJ), 100 U/mL penicillin, and 100 μg/mL streptomycin.

**Generation of reporter constructs.** The HBZ promoter, 3′LTR300[20], and 5′LTR were amplified from ED cells. The NFR was amplified from ED cells and the NFR mutant was generated by gBlocks® Gene Fragments (Integrated DNA Technologies, Coralville, IA). Using XhoI and HindIII restriction sites, each promoter construct was inserted into pGL4-basic (Promega, Madison, WI) which includes the luciferase reporter gene. The NFR was inserted into pGL4-3′LTR300 or pGL4-5′LTR using BamHI or KpnI restriction sites while the NFR mutant was inserted into pGL4-3′LTR300 using the BamHI restriction site. NFR-CTCF fragments were cloned into pGL4-3′LTR300 or pGL4-5′LTR by NEBuilder HiFi DNA Assembly Master Mix (New England Biolabs, Ipswich, MA). Primers associated with each construct and the NFR mutant are listed in Supplementary Table 3.

**mRNA-seq and qRT-PCR.** RNA was extracted using the RNeasy Mini Kit (Qiagen, Hilden, Germany) according to the manufacturer's instructions and treated with DNase. For mRNA-seq, mRNA libraries were prepared using NEBNext® Ultra™ II Directional RNA Library Prep Kit for Illumina® Multiplex Oligos for Illumina (New England Biolabs) according to the manufacturer's instructions. Libraries were run as 75-cycle-single end reads on a NextSeq 550 (Illumina, San Diego, CA) using a high-output flow cell. cDNA was synthesized using ReverTra Ace® qPCR RT Master Mix (Toyobo, Osaka, Japan) according to the manufacturer's instructions. qPCR was performed using Thunderbird SYBR qPCR mix (Toyobo) and run on an Applied Biosystems® StepOnePlus™ Real-Time PCR System (Thermo Fisher Scientific); primers used are listed in Supplementary Table 4.

**Preparation and culture of HTLV-1-infected cells in vitro.** 293 T cells were transfected with a wt or enhancer-mutated HTLV-1 molecular clone[48] by polyethylenimine (PEI) and then irradiated with 30 Gy. The irradiated 293 T cells were co-cultured with JET cells for 3 days[47], after which tdTomato-positive cells were sorted by FACS Aria™ (Becton, Dickinson and Company, Franklin Lakes, NJ), and cultured in RPMI supplemented with 10% FBS, 100 U/mL penicillin, and 100 μg/mL streptomycin for 2 weeks.

**Proviral load (PVL) measurement.** We estimated the number of infected cells by quantifying the copy number of the *tax* gene normalized to the copy number of the *ALB* gene by using digital droplet PCR as previously described but with minor modifications[15]. PVL was calculated as follows, PVL (%) = [(copy number of *tax*)/(copy number of *albumin*)/2] × 100. Primer sequences are listed in Supplementary Table 4.

**HTLV-1 DNA-capture-seq.** HTLV-1 DNA-capture-seq was performed as previously described[15] with minor modifications. Briefly, 1 μg genomic DNA was fragmented by sonication using a Picoruptor (Diagenode s.a., Liège, Belgium) to produce 300–500-bp fragments. The DNA library was generated using a NEBNext Ultra II DNA Library Prep Kit for Illumina and Multiplex Oligos for Illumina (New England Biolabs). DNA-seq libraries were used for HTLV-1 sequence enrichment with HTLV-1 specific probes, after which enriched libraries were amplified by additional PCR. Enriched libraries were quantified using P5 and P7 primers and then sequenced via Illumina MiSeq or NextSeq.

**MNase assay and MNase-seq.** Cells ($1.0 \times 10^6$ for cell lines or $2.0 \times 10^6$ for patient PBMCs) were lysed using cell lysis buffer (0.05% Triton X-100, 2 mM PMSF, 5 mM sodium butyrate, 100× protease inhibitor cocktail) or PBMC lysis buffer (10 mM Tris-HCl pH 7.4, 10 mM NaCl, 3 mM MgCl₂, 0.1% Nonidet-P40). Extracted nuclei were digested by MNase (TaKaRa Bio, Kusatsu, Japan) for 5–20 minutes at 37 °C after which the reaction was stopped by the addition of 20 mM ethylenediaminetetraacetic acid (EDTA). After deproteination with proteinase K solution (Nacalai Tesque, Kyoto, Japan), MNase digestion samples were purified using a PCR Purification Kit (Qiagen). We confirmed constant MNase treatment among different samples by gel electrophoresis after MNase digestion. MNase-digested DNA and input DNA were measured either by QX200 droplet digital PCR system (BIO-RAD, Hercules, CA) or by next-generation sequencing (NGS). Primer sequences for ddPCR are shown in Supplementary Table 6. For NGS, MNase-seq libraries were prepared by the NEBNext Ultra II DNA Library Prep Kit for Illumina and Multiplex Oligos for Illumina (New England Biolabs), after which the efficiency was quantified using P5 and P7 primers and then sequenced using Illumina MiSeq. We prepared fragmented DNAs by sonication using a Picoruptor (Diagenode s.a., Liège, Belgium) and use them as input DNAs. The degree of nucleosome freeness which was calculated as the MNase-seq value normalized to the input DNA value.

**ChIP-seq.** ChIP assays were performed using the SimpleChIP® Enzymatic Chromatin IP Kit (Cell Signaling Technology, Danvers, MA) according to the manufacturer's instructions. Briefly, cells ($4 \times 10^6$) were fixed in 1% formaldehyde for 10 min at room temperature, quenched in glycine solution, and washed in ice-cold PBS. Nuclei were extracted by lysis buffer (buffer A) and then samples were digested by MNase for 20 min at 37 °C and sonicated for 30 s on and 30 s off for 5–8 min using Bioruptor UCD-300 (Tosyodenki, Kanagawa, Japan) to break the nuclear membrane. Extracted chromatin was immunoprecipitated using anti-H3K27Ac (#07-360; Millipore, Burlington, MA), H3K4me1 (#ab8895; Abcam, UK, England), H3K4me2 (#ab7766, Abcam), SRF (#5147; Cell Signaling Technology), and ELK-1 (#ab32106; Abcam) antibodies. All antibodies were used at a 1:250 dilution. ChIP sample libraries were prepared by NEBNext Ultra II DNA Library Prep Kit for Illumina and Multiplex Oligos for Illumina (New England Biolabs), after which the efficiency was quantified using P5 and P7 primers and sequenced

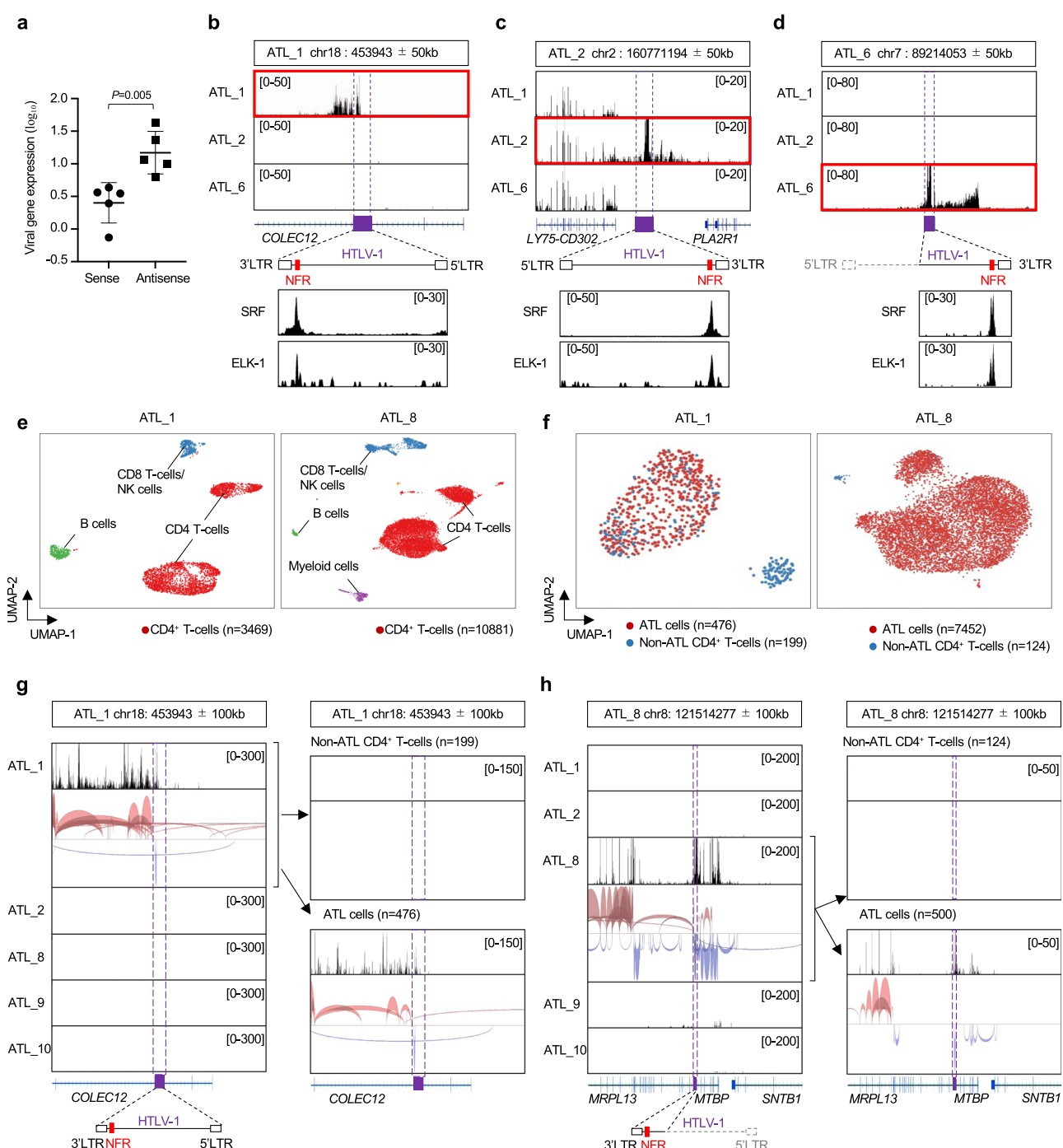

**Fig. 7 Transcriptional characterization of the provirus and the flanking host genomes in freshly isolated PBMCs from infected individuals. a** Expression level of sense or antisense proviral transcripts in fresh PBMCs from five ATL patient samples (ATL_1, ATL_2, ATL_5, ATL_6, ATL_7). Data were shown as transcripts per million from bulk mRNA-seq for each case. $n = 5$ biologically independent samples, mean ± SD. $P$ values are calculated by a two-sided Student's $t$-test. **b**–**d** Visualization of mRNA-seq data of three ATL cases (**b** ATL_1, **c** ATL_2, and **d** ATL_6) around each proviral IS. Host genes near the IS and direction of HTLV-1 provirus are indicated under mRNA-seq signal images. SRF/ELK-1 ChIP-seq results without HTLV-1 DNA-capture are also shown for each ATL sample. The ATL sample with HTLV-1 IS in the region is highlighted with a red square. **e** scRNA-seq data of PBMCs from the indicated ATL cases. Cell clusters were annotated by examining the expression pattern of the marker gene for each PBMC subset as described in Methods. **f** Figures show the distribution of ATL cells and non-ATL CD4+ T-cells for the two ATL cases in (**e**). ATL cells are defined as the T-cells with the most abundant T-cell receptor (TCR). **g**, **h** Local transcriptome including viral integration site are visualized by IGV. We obtained scRNA-seq data from five ATL cases. The data shown were regions with viral IS of **g** ATL_1 and **h** ATL_8 respectively. Data from all CD4+ T-cells are shown in the left panel. CD4+ T-cells are further divided into non-ATL CD4+ T-cells (right, upper panel) and ATL cells (right, lower panel) based on their TCRs.

using Illumina MiSeq or NextSeq. To increase the detection sensitivity, we analyzed with the HTLV-1 DNA-capture method as reported previously[15]. Briefly, after library preparation, we mixed libraries and biotinylated DNA probes for the whole HTLV-1 provirus, and then HTLV-1 sequences were enriched by streptavidin. The enriched libraries were analyzed by Illumina MiSeq or NextSeq.

**Luciferase reporter assays**. Jurkat cells ($2 \times 10^5$) were harvested 24 h after transfection with 1 µg of each reporter construct, using 2 µl of Turbofect Transfection Reagent (Thermo Fisher Scientific). Luciferase assays were then performed using the Dual-Glo Luciferase Assay System (Promega) according to the manufacturer's instructions, and luminescence was detected using GloMax® 20/20 Luminometer (Promega).

**NET-CAGE**. Nascent RNAs were extracted from the nuclei of ED cells and TBX-4B cells following the previously described[23]. NET-CAGE libraries were generated using the CAGE library preparation kit (K.K. DNAFORM, Yokohama, Japan) according to the manufacturer's instructions. Briefly, cDNA was synthesized from 5 µg nascent RNAs. The 5′cap-structures of nascent RNAs were labeled by 4 µl of 10 mM biotin hydrazide for the cap-trapping step. After removing the Remaining RNA fragments without 5'cap structure by RNaseONE enzyme, enriched cDNA by cap-trapping was used for linker ligation and library generation. NET-CAGE Libraries were quantified by qPCR and sequenced using Illumina NextSeq.

**EMSA (electrophoretic mobility shift assay)**. 293 T cells ($2 \times 10^6$) were harvested 24 h after transfection with 2 µg of pcDNA3-myc-SRF[49] and 2 µg of pCGN-ELK-1 (Addgene, Watertown, MA) using 16 µl of HilyMax (Dojindo Laboratories, Kumamoto, Japan). Cell lysates were extracted in 500 µl of cell lysis buffer (10 mM Tris-HCl pH 8.0, 60 mM KCl, 1 mM EDTA, 1 mM DTT, 100 µM PMSF, 0.1% NP-40) with 5 min incubation on ice. After cell lysis, nuclear lysates were extracted in 100 µl of nuclear extraction buffer (20 mM Tris-HCl pH 8.0, 420 mM NaCl, 1.5 mM MgCl2, 0.2 mM EDTA, 25 % glycerol) with 10 min incubation on ice and then samples were sonicated for 20 on and 30 s off for 17 min using Bioruptor UCD-300 (Tosyodenki) to break overfilled DNA. EMSA was performed with the 1 µl of extracted nuclear lysates, biotin-labeled NFR-wt probe, and NFR-wt or mut unlabeled probes using Perfect NT Gel which is a 3–12% gradient polyacrylamide gel (#NTH-5X5HP; DRC, Tokyo, Japan) and the LightShift Chemiluminescent EMSA Kit (#20148; Thermo Fisher Scientific) according to the manufacturer's instructions. Nuclear lysates were mixed with 50 fmol biotin-labeled probes and 1 µg each of the anti-SRF (#5147; Cell Signaling Technology) and anti-ELK-1 (#ab32106; Abcam) antibodies. For the competition assay, NFR-wt or -mut unlabeled competitor probes (5, 10, and 15 pmol) were added in the mixture of nuclear lysates and biotin-labeled NFR-wt probe. Probe sequences are listed in Supplementary Table 5.

**p19 ELISA**. 293 T cells ($2 \times 10^5$) were transfected with HTLV-1-wt or mut molecular clone (0.25, 0.5, and 1 µg) using 3 µl of HilyMax (Dojindo Laboratories). After 24 h, the supernatants were collected and measured p19 presence by RETROtek HTLV p19 Antigen ELISA (ZeptoMetrix Corporation, Buffalo, NY) following the manufacturer's instruction.

**Apoptosis analysis**. JET cells infected with HTLV-1-wt or mut molecular clone were stimulated with 100 ng/ml PMA and 2 µM Ionomycin and incubated for 24 h. After incubation, apoptotic cells were stained with annexin V by MEBCYTO® Apoptosis Kit (MBL, Nagoya, Japan) and detected by flow cytometry using BD FACSVerse™ (Becton, Dickinson and Company). Flow cytometry data were analyzed using FlowJo™ (Becton, Dickinson and Company). Gating strategies for annexin V+ cells were shown in Supplementary Fig. 8

**CRISPR/Cas9 mutagenesis**. Guide sequences were designed with both edges of NFR in target and cloned into the pX330-U6-Chimeric BB-CBh-hSpCas9 plasmid (pX330; Addgene, 42230) as previously described[50]. The oligonucleotides for constructing guide sequence were listed in Supplementary Table 7. HTLV-1-wt-infected JET clone ($2 \times 10^6$) was co-transfected with each 3 µg of two pX330 plasmids for each NFR edge, 1.5 µg of an expression vector with puromycin resistance gene and 3 µg of NFR_mut_3 cassette plasmid for HDR by electroporation using NEPA21 (NEPAGENE, Ichikawa, Japan) and 2 mm gap cuvette (EC-002S, NEPAGENE). The electroporation program was following; 275 V, 1 ms, six times, and a 50 ms interval for poring pulse, and 20 V, 50 ms, three times, and a 50 ms interval for transfer pulse. Transfected cells were selected by 1 µg/mL puromycin with 24 h incubation. After transfected cells were recovered and damages for 3 days, limiting dilution was performed to get a single clone. CRISPR/Cas9 mediated mutant clone was confirmed the sequence which converted wt to mut by Sanger sequencing.

**Single-cell RNA-seq analysis**. Single-cell data acquisition was performed in a previous study[51] and the sequencing data were obtained from the European Nucleotide Archive (ENA) (https://www.ebi.ac.uk/ena/browser/home) with the

following accession numbers for each sample: ATL_1, ERX6294562; ATL_2, ERX6294563; ATL_8, ERX6294567; ATL_9, ERX6294566; and ATL_10, ERX6294565. Cell Ranger Single-Cell Software Suite (v3.1.0, 10x Genomics, Pleasanton, CA) was used to perform sequence alignment against a modified hg38 human reference genome which contains the HTLV-1 genome (Genbank accession no. AB513134) as a separate chromosome. The resulting barcode matrix was imported into R (v4.0.3) and analyzed using Seurat (v4.0) according to the vignette on Seurat's webpage available here (https://satijalab.org/seurat/articles/pbmc3k_tutorial.html). Clusters were annotated based on examination of known marker genes for each PBMC subsets. In this case, CD4 T-cells, CD8/NK cells, B cells and myeloid cells are CD3D+CD4+, CD3D+CD8A+NKG7+, CD79A+CD19+, and CD14+FCGR3A+ respectively[52]. T-cell clones are identified based on TCR information with the ATL clone defined as the most expanded CD4+ T-cell clone in the sample.

**Western blot**. We generated wt- or mutated-tax (tagged c-Myc) pcDNA3.1(-)-c-Myc vectors based on pcDNA3.1(-) (Invitrogen, Waltham, MA). 293 T cells ($2 \times 10^6$) were harvested 24 h after transfection with 6 µg of wt- or mutated-tax (tagged c-Myc) pcDNA3.1(-)-c-Myc vectors using 16 µl of HilyMax (Dojindo Laboratories). After cell lysis in 200 µl of RIPA buffer (50 mM Tris-HCl pH 7.4, 150 mM NaCl, 0.1% SDS, 1% Triton X-100, 1% sodium deoxycholate, 1 mM EDTA) with protease inhibitor cocktail and phosphatase inhibitor cocktail, 30 µg of cell proteins were separated on the precast gel for SDS-PAGE (197-15011; FUJIFILM, Osaka, Japan). Separated proteins were blotted onto polyvinylidene difluoride membrane (5400412; ATTO, Tokyo, Japan). About 0.1 µg/ml anti-Myc antibody (M192-3; MBL, Tokyo, Japan) and 0.1 µg/ml anti-Actin (C-2) antibody (sc-8432; Santa Cruz Biotechnology, Dallas, TX) were used as primary antibodies staining. secondary reactions with HRP were performed using Pierce™ Fast Western Blot Kit, ECL Substrate (35055; Thermo Fisher Scientific) according to the manufacturer's instruction. Chemiluminescent detection was performed on a ChemiDoc™ Touch Imaging system (BIO-RAD).

**Bioinformatic analysis**. Prediction of transcription factors which bind to the NFR sequence was performed using TFBIND (https://tfbind.hgc.jp). Analysis of next-generation sequencing data was performed as follows. First, the quality of raw FASTQ files were checked with FastQC (v 0.10.0) followed by adapter trimming and removal of poor-quality reads using cutadapt (v 1.18) and PRINSEQ (v 0.20.4) respectively. For ChIP-seq analysis, peak calling was performed using MACS (v 1.4.2) as described previously[11]. For the HTLV-1-DNA-seq data, alignment to reference genome was performed using BWA (v 0.7.12) followed by viral integration site and clonal abundance analysis using samtools (v 1.11), picard (v 2.0.1), and in-house perl scripts as we previously reported[15]. RefSeq gene data were obtained from UCSC tables (https://genome.ucsc.edu/). The relationship between viral integration site and host genes or epigenetic microenvironment were analyzed using the R package hiAnnotator (http://github.com/malnirav/hiAnnotator) as described previously[53]. Gene expression for bulk RNA-seq data of cell lines (ED and TBX-4B), parent (JET), wild type, and mutated clones (see Supplementary Table 2) was quantified using kallisto[54] and exported into R for differential expression analysis using the R package DESeq2[55] (https://github.com/mikelove/DESeq2). Data were filtered to remove genes with low counts (<10) followed by log-transformation and visualization on a PCA plot. The same RNA-seq data were also aligned to a reference genome using STAR (v 2.7.3) andfor visualization on IGV (v 2.8.0). Data for doxycycline-induced Tax- or HBZ-expressing Jurkat cells were obtained from SRA1049749[27]. Fold change of gene expression by *tax* or *HBZ* was calculated with respect to the non-induced cells (i.e. Doxycycline-induced HBZ-Jurkat against non-induced HBZ-Jurkat).

**Statistical analysis**. Data were analyzed using a chi-squared test with GraphPad Prism 7 software (GraphPad Software Inc., La Jolla, CA) unless otherwise stated. Statistical significance was defined as $P < 0.05$.

**Reporting summary**. Further information on research design is available in the Nature Research Reporting Summary linked to this article.

## Data availability
All NGS sequencing data generated in this study have been deposited in the DDBJ Sequence Read Archive (DRA), which is associated with DNA DataBank of Japan (DDBJ) under accession code DRA013478 (https://ddbj.nig.ac.jp/resource/sra-submission/DRA013478). Processed data have been deposited in the DRA under the accession number DRA013591 (https://ddbj.nig.ac.jp/resource/sra-submission/DRA013591) and DRA013588 (https://ddbj.nig.ac.jp/resource/sra-submission/DRA013588). Peak call data of ChIP-seq have been deposited in the Genomic Expression Archive (GEA), which is associated with DDBJ under an accession number E-GEAD-481. Raw experimental data such as luciferase assay, EMSA, ELISA, qRT-PCR, PVL measurement, cell apoptosis assay, MNase assay, PCA plots, TPM of RNA-seq, cell clustering assay of scRNA-seq, integration site distribution analysis, western blots are available from Source Data file.

## Code availability

The source code to reproduce our analysis has been uploaded to GitHub and linked to Zenodo: https://doi.org/10.5281/zenodo.6361824.

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

## Acknowledgements

We would like to thank M. Miura for providing the R program to perform quality checks of the index reads using the R program, M. Nakao for providing the SRF expression vector, N. Misawa, S. Nagaoka, and K. Sato for technical support and valuable discussion. We are also grateful to CRM. Bangham, S. Hino, and M. Ono for their critical reading of the manuscript. This study was supported by grants from the Japan Society for the Promotion of Science (JSPS) KAKENHI (JP20H03724, and JP18KK0230 to Y.S., 16KK0206 and JP18K16122 to H.K., JP18K08437 and JP18KK0452 to P.M., JP20K22783 and JP21K08494 to K.S.; JP21K15454 to M.M.) and Japan Agency for Medical Research and Development (AMED) (JP21jm0210074, JP21wm0325015, JP19fm0208012, JP21fk0410023, and JP21wm03250152to Y.S.) the Grant for Joint Research Project of the Institute of Medical Science, the University of Tokyo to Y.S., the grant from Kumamoto University Excellent Research Projects to Y.S., JST MIRAI (18077147) to Y.S., the program of the Joint Usage/Research Center for Developmental Medicine, Inter-University Research Network for Trans-Omics Medicine, Institute of Molecular Embryology and Genetics, Kumamoto University to Y.S. and Kumamoto University Fellowship for Excellent Graduate Students to M.M. The funders had no role in study design, data collection, data interpretation, or the discussion regarding submission for publication.

## Author contributions

M.M acquired funding for the project, designed and performed almost experiments, including epigenetic profiles, gene expression profiles, wt or mutant HTLV-1 profiles, functional analysis of the target proviral region, flow cytometry, genome editing, bioinformatic analysis, data curation, and wrote the paper. T.U. and K.M. established wt or mutant HTLV-1 infected clones for analysis of provirus profiles. K.S. acquired funding for the project, performed a functional analysis of the target proviral region, and wrote the paper. B.J.Y.T. performed a bioinformatic analysis of scRNA-seq and wrote the paper. A.R., K.U., and S.I. generated DNA libraries, performed DNA-captured-seq, and proviral load measurements. P.M. acquired funding for the project and performed NET-CAGE. H.K. acquired funding for the project and performed HTLV-1 integration-site (IS) analysis and IS characterization. M.T., K.N., and A.U. provided clinical samples. S.N. produced data for characterization of wt or mutant HTLV-1 infected cells. H.H. and J.F. supervised the project. Y.S. conceived and supervised the project, acquired funding for the project, performed data curation and bioinformatic analysis, and wrote the paper. All authors discussed the results and commented on the manuscript.

## Competing interests

The authors declare no competing interests.
