## [Peer Review File · Nature Communications]

Reviewer comments, first round -

Reviewers' Comments:

Reviewer #1:

Remarks to the Author:

The study by Matsuo and colleagues describes a previously unidentified viral enhancer in viral genome of the HTLV-1, the causative agent of Adult T-cell leukemia (ATL). The authors show binding of transcription factors SRF and ELK-1 to this enhancer region in ATL-derived cell lines and freshly isolated leukemic cells. Using a cloned provirus containing mutations in the SRF- and ELK-1-binding sites they observe decreased chromatin openness at the viral enhancer, reduced viral gene transcription and changes in host gene transcription near the viral integration site in mutant virus infected cells. scRNA-seq of ATL cells revealed aberrant host genome transcription nearby integration sites of full-length and 5'LTR-deleted defective proviruses containing this enhancer.

This is an interesting study that reveals and explores a new regulatory element in the HTLV-1 proviral genome. While the work is attractive and of interest for the HTLV-1 research community, it comes with a number of questions and concerns.

Main comments:

Title:

What do the authors mean by "hidden"? We agree the enhancer region was unidentified until now, but is it really hidden?

Replace retroviral by HTLV-1 (enhancers in other retroviruses have been identified previously, including in ERVs)

By viral DNA capture seq approach: basically, the viral enhancer element and its main characteristics (chromatin marks, TF binding, ...) were revealed via classical methods including CHIP-seq, NET-CAGE, RNA-seq, scRNA-seq etc ... with the help of the viral DNA capture method previously developed by the authors. Viral DNA capture-seq was used in combination with these methods to target and enrich viral sequences in a subset of experiments.

Novelty:

While the viral enhancer is novel in the context of HTLV-1, the presence of viral enhancers within retroviral genomes is not new. It was previously shown that such enhancers influence host gene expression in the vicinity of their genomic location. This is also the case for endogenous retroviral elements as demonstrated by recent work by Ozgen and colleagues (Nat Comm 2020). This work reveals that ERV-derived regulatory sequences are a source of enhancers that may be exploited by cancer cells to help drive tumour heterogeneity and evolution.

Line 109

We further asked if the NFR is also observed in in vivo samples in addition to the in vitro cell lines by analyzing peripheral mononuclear cells (PBMCs) freshly isolated from ATL patients and an asymptomatic carrier.

ATL patients and AC: how were they selected?

Patient primary samples and their characteristics are not described in the manuscript while several ATLs and AC samples are used in the study. ATLs are defined as "smoldering, acute, chronic" in Fig 1. In Figs 5 and 6, ATLs have numbers ranging from 1 to 8. Are some of these patient samples identical? Providing a table of patient samples used in the study with their clinical and molecular characteristics will increase clarity.

Fig 1: degree of nucleosome freeness, how is the total input DNA-seq value computed?

Can the authors comment on the observation that in vivo patient samples (AC and ATLs) have

similar MNase-seq profiles than the sense RNA-producing cell line (TBX-4B)? In both patient types, and more specifically in ATL cases, sense transcription is barely detected (this is mentioned in the introduction and repeated many times in the manuscript). Why is this not the case for NFRs? Is there a bias generated by the method? How were these samples processed?

Regarding the insulator region, the NFR at this location is not significantly more represented than other regions all over the provirus, except for ED. Same question as above.

In IGV figures, explain the numbers between brackets (scale, coverage) in figure legends. Readers not familiar with IGV representations may get lost.

CHIP-seq and Fig 2: can the authors comment on the resolution of their CHIP-seq experiments? The CHIP-seq data and images are not fully convincing. There is a broad coverage of Chip-seq reads. How was the specific attribution to the NFR defined? It seems that the reads are covering the entire HBZ transcript region, or did I miss anything?

Chromatin immunoprecipitation sequencing (CHIP-seq) signals of enhancer-related histone modifications, including H3K27Ac, H3K4me1, and H3K4me2, were high around the NFR in ED cells (Figure 2e, upper panel).

What is meant by around?

Luciferase experiments: how does luciferase activity change when Tax expression is added? The in vivo data in Fig 1 (ATL, AC) show sense-related NFR and possibly some positive strand expression, how would tax influence promoter activity with or without the enhancer region on both the 5' and 3' promoter (LTR)? Would this better reflect the situation in vivo, where despite a low sense activity, CTL responses to tax reveal a persistent tax expression.

The legend to Fig 2 is unclear: what do the authors mean by the nucleosome region? NFR? Nucleosome-free?

Fig. 2 The nucleosome region harbors enhancer-related histone modifications and produces enhancer RNAs

Fig 2e: the NET-CAGE experiment generates interesting results, however the data could be commented in greater detail. It would also be useful to see this method applied to TBX-4B. Why was it not performed? Again, in vivo, we also expect finding proviruses that are not impaired in sense transcription by mutation or epigenetic silencing, not only "ED-like" proviruses. Performing NET-CAGE with patient samples (this was also done in the case of sensitive CHIP-seq, below) would be an asset and biologically relevant.

Line 171

The NFR region we identified in this study is ~160 bp in length

How this was this determined?

Fig 3:

a The prediction of transcription factor binding to the NFR was performed by using TFBIND (<http://tfbind.hgc.jp/>). Candidate transcription factor binding sites are shown.

Not clear what the numbers 8000 and 8160 represent? How do they relate to the indication of 7100 earlier in the manuscript?

Why were SRF and ELK1 chosen among all candidates? Were they tested in HTLV-1 viral capture CHIP-seq described by the authors as being very sensitive?

How was this performed? If I am correct, the CHIP-seq capture-seq combination method is not described.

Line 225

(The) SRF and ELK-1 plays a critical role in HTLV-1 enhancer function

Fig S1a Tax protein levels in nuclear lysates of cells transfected with wt or enhancer region-mutated Tax-expression vectors.

Was this tested with expression vectors, or using full-length proviruses? How was this experiment performed? Is this described in Methods?

Suppl Fig 1a. Where is SFig 1b and potentially c, d, ...?

After quantifying viral gene expression in the transiently transfected cells and viral production in the culture supernatant (Figure 4a), we found a marginal decrease of p19 production in the supernatant of mut plasmid-transfected cells; however, there was no statistically significant difference (Figure 4b). Nevertheless, there was a significant reduction in tax and HBZ expression at the mRNA level (Figure 4c).

Tax expression has an impact on 5' LTR promoter activity. The authors observed decreased tax expression, but no significant change in p19 expression. Could they comment on this?

JET cell infection experiment: if tax is decreased in mutant proviruses, why is it not the case in infected cells?

Line 245

The proviral load of HTLV-1-mut-transfected JET cells was lower than that of HTLV-1-wt-transfected ones (Figure 4f).

It is not clear if the authors produced both transfected and infected JET cells. Is 'transfected' correct, or are these cells infected? Please clarify.

How do the authors explain reduced PVL and is this important? PVLs in Fig 4 suggest multiple IS per cell (>2) for wt. Since the authors examine bulk data, how could they determine the distribution across clones? Cells are sorted according to the integrated reporter plasmid (tax response element and fluorescence) activity, suggesting they rely again on tax expression that is said decreased in mut proviruses.

Distribution of viral IS was not so different between the WT and mutant HTLV-1-infected JET cells in terms of the relationship with the host gene and epigenetic environment Figure. S2a and S2b).

Wouldn't the opposite be expected if the impact of NFR on neighboring genomic regions provides an advantage to the cell? If the presence of the viral enhancer is critical, we may expect seeing the selection of cells that have increased fitness, even in vitro?

Fig 4 k MNase assay of in JET cells infected with HTLV-1-wt (above) or mut (below).

Fig 5: in JET clones wt and mut

Line 316

The proportion of upregulated genes in JET clones infected with HTLV-1-wt was significantly higher than those in mutant HTLV-1 clones ($P < 0.01$; Figure 5d).

Is this related to overall expression or is this due to transcripts in the neighborhood of HTLV-1 ISs?

Line 321

We then used CRISPR/Cas9 to introduce the mutation that abrogated SRF-ELK-1 binding to the enhancer region (Figure 3d and 3e, Figure 4g) of a clone infected with wt-HTLV-1. SRF/ELK-1 ChIP-seq peaks in wt-HTLV-1-infected cells were abolished in the CRISPR-mutated cells, thereby reducing proviral transcription both at sense and antisense direction (Figure 5e and 5f) and chromatin openness in the enhancer region (Figure 5g).

How many IS were found in this clone and have all (if multiple) corresponding proviruses been edited by Crispr/Cas9?

Fig 5: limiting dilution

Does limiting dilution introduce bias? Fit clones will have a significantly higher chance to survive the cloning procedure.

Was there only one clone selected for wt and one clone for the mut condition? How many IS?

Previous data suggest the number of IS per clone is > 1 . Did the authors in purpose select a clone with only 1 IS?

The mut provirus Jurkat clone has an IS in a gene that is expressed in parental cells (mRNA, RNA-seq) while the wt provirus has an IS in a gene that does not show expression (parental RNA-seq). Is it correct to compare these two cases? We suggest examining several clones with ISs located in both expressed and non-expressed genomic regions for both situations (wt and mut). In summary, this is a very interesting experiment, however, it is statistically unsound. A larger number of clones should be examined, not just one nice example of each condition (wt and mut). This would significantly strengthen the authors' conclusions.

Crispr/Cas mut clone: which wt clone was used for this experiment? Same clone as the one illustrated above (Fig. 5)?

Primary cell studies:

Line 359

We further investigated the effect of HTLV-1 integration on viral and host genomes by performing mRNA-seq analysis using freshly isolated PBMCs from five ATL cases.

How were ATL cases selected, and how do they relate, if they do, to the primary samples shown in Fig. 1 (see comments above)?

Line 367

More importantly, there were clear peaks of SRF and ELK-1 ChIP-seq signals in integrated proviruses, indicating SRF and ELK-1 play a role in the transcriptional regulation (Figure 6b-6d).

Was this explored by what the authors called sensitive Capture-mediated CHIP-seq?

Line 380

Furthermore, there was a significant increase of the local transcriptome in the ATL clone but not in non-ATL CD4+ T cell clones (Figure 6g and 6h, Figure S4a-4c, right panels).

How many single cells were examined for each ATL?

In Fig 6 g and h (scRNA-seq data in IGV), does each IGV graph represent a single cell? How many did the authors screen? Are none of the genes in the vicinity of the IS in ATL 1 or ATL 6 expressed in uninfected T-cells or HTLV-1 infected non ATL clones (COLEC12, MRPL13, MTBP)? Same comment, how many non ATL or uninfected T-cells did the authors screen?

What is the percentage of single cells belonging to the ATL clone (defined by matched TCR) that showed a transcriptome profile identical to that illustrated in Fig 6 in the IS vicinity? Is it expected that all ATL cells express these local transcripts at a given time?

Did the authors detect HBZ transcription in all single cells? If not, in what percentage of cells and how does the level of HBZ transcript expression compare to the distribution of host cell transcripts?

It should be possible to examine the local transcriptome in single cells belonging to other smaller clones since the authors claim that the viral capture-seq method can identify the majority of IS including the non ATL integration sites. Do the authors find transcriptional changes in the vicinity of these ISs by scRNA-seq? Is modification of the transcriptome around the IS a specific feature of the transformed ATL clone? Is it related to tumor progression?

Generally speaking, the in vivo experiments described in the last section of the manuscript are of

substantial interest, but the experiments are poorly documented especially the scRNA-seq study. Both the experimental design (including information about the number of cells examined) and the bioinformatics analysis (i.e. analysis of the spliced transcripts in the vicinity of the IS in single cell data, provirus-driven transcript modifications) require more detailed explanation to improve clarity.

Do the authors expect being able to identify all or the majority of transcripts that have a junction between the virus and the host genome using 10X Genomics scRNA-seq? Is the standard configuration of 10X Genomics (91-98 bp reads) appropriate for the detection of these transcripts at the single cell level?

Is the viral enhancer capable of driving host gene transcription on its own, thus in combination with host gene promoter(s) and regulatory regions but without the implication of the viral promoter, or does it require the HBZ promoter to exert its function?

The impact of defective HTLV-1 proviruses on the local host transcriptome in ATLs carrying a 5'LTR deleted provirus has been previously reported (Kataoka, 2015, Rosewick, 2017).

Minors comments:

Several errors in Methods, Figure legends, Discussion. Examples in Fig 1: Stranded proviral transcriptome (s?) are ...; in Methods: ED, 293T, Jurkat, and JET cells infected (with) WT or mutant HTLV-1 molecular clones; in Discussion: Consistent with this notion, the intragenic viral enhancer and insulator are maintained even in defective type proviruses that is (are) observed (in) 20-30% of ATL cells; thereby upregulates cancer-related genes near ISs and might contribute (contribute) to the selection of a specific infected cell for clonal expansion

Reviewer #2:

Remarks to the Author:

The manuscript by Matsuo et al. aimed at identifying nucleosome-free DNA regions (NFR) in the HTLV-1 provirus sequence and investigating their impact on HTLV transcriptional activity, both in the sense and antisense directions. Using micrococcal nuclease digestion coupled with enrichment of HTLV DNA sequences, they identified such a NFR region at the 3' end of the proviral sequence, between an insulator sequence and the 3'LTR promoter element. They showed that this NFR sequence was indeed able to enhance the 3' LTR-mediated transcriptional activity, but not the 5' LTR one. They further identified SRF and ELF-1 as transcription factors binding to that NFR and regulating its activity (via ChIP-Seq analyses, EMSA and mutagenesis experiments). Finally, they provided transcriptomic data showing the impact of HTLV provirus insertion, complete or defective, on transcription activity in integration site surrounding region (readthrough).

Overall, the study provides a plethora of experiments confirming the identification and relevance of this novel enhancer element. However, a few issues need to be answered to ensure the absence of bias in the analysis and the pertinence of the conclusions. Furthermore, the manuscript might be improved with additional information to facilitate reading and understanding.

Major issues :

- One of the aim is to identify a putative role of the NFR enhancer on the sense and antisense transcription. Although there is no doubt that this NFR is playing a role for antisense transcription, NFR impact on sense transcription seems minimal in the luciferase assay (Figure 2c). However, I am wondering if in this assay the design could be biased as the enhancer is used with the complement 3'U5 region only vs the 5' whole LTR on the other hand. Furthermore, it may be useful to include in Figure 2c constructs that also include the insulator before and after the enhancer as controls, to further mimic the proviral sequence. On another hand, could you please clarify the impact of this NFR on the sense transcription as results from figure S1 and figure 4i are confusing, and potentially contradictory.
- The data obtained from ChIP-Seq for histone modification marks (Figure 2e) are not very conclusive and the authors should be careful interpreting these data (lines 137-139). Indeed "high" levels of histone marks "around" the NFR cannot be stated in my opinion.
- Analysis of the NFR region provided multiple putative transcription factors binding to it. It is not

clear why the authors decided to investigate only SRF and ELK-1. Indeed, the other candidates should be investigated as well as otherwise the whole study can be misleading. Although there is no doubt that SRF and ELK-1 bind to this NRF enhancer, there might not be the instigators of the transcription seen. Indeed, it could well be that Stat-3 for instance is also binding and that the mutations generated also prevent its binding and activity.

- In my opinion, the authors speculate on the function of the NRF enhancer on sense/antisense transcription and cooperation with CTCF (lines 416-419 and 434-435), without having been able to completely and formally demonstrate it,

- Title suggestion : Identification and characterization of a novel HTLV-1 enhancer by a viral DNA-Capture-Seq approach

- The introduction should provide more background information, especially between the different forms of ATL (acute, chronic, smoldering).

- The methods are very superficial and do absolutely not allow to reproduce the experiments presented as such. Cell, plasmid, or compound concentrations are missing as well as durations or volumes; for instance how much MNase was added, antibody dilution at 1:250, transfection conditions, amount of plasmid transfected, puromycin concentration and days of selection, etc. A section about plasmids and their description might be introduced. For instance, when performing ChIP-Seq experiments, is DNA-capture performed after or before the ChIP ? When assigning reads to the provirus sequence, how did you handle the assignment to 5' vs 3'LTR regions ?

- Figure legends are incomplete and often do not contain all the descriptive information. For instance, in the IGV viewers, please specify what are the numbers in square brackets. Figure 2f, what is the color coding, and the upper and lower graphs in sense and then antisense transcripts ?

- Figure 1c : please add a scale with nucleotide position on top of the provirus sequence, indicate the start and end of each region (LTR; insulator, NRF).

- Lines 133-135 about Figure 2d : please stick to the facts as here data are not significant, so the text cannot state that there is a marginal increase.

- Figure 3d: please provide the complete and exact sequence as supplementary information.

- Figure 4h : please provide a supplementary table with the list of integration sites and features.

- The rationale for analyzing CTCF (lines 318-320 and figure 5d) is not clear, please provide more information.

- ATL cells from patients (Figure S3) : please provide additional information regarding the ATL degree (acute, smoldering, ...)

- Figure 6a : please include all patients, why only ATL-1 to 5 ?

- Figure 6e : please explain in the methods which data did you use to annotate PBMC subset.

Minor issues :

- Lines 33-35 : the sentence is confusing, as it indicates that antisense transcription (thus from the 3'LTR) can be active event if the 5'LTR sequence is defective. I don't understand this link as antisense transcription should not be dependent from a functional sense transcription to my knowledge.

- Line 113 : please rephrase "... in naturally virus-infected cells from individuals).

- Figure 2e : please add as well the orange shadow for the picture.

- Line 152 : please modify "The nucleosome-free region ..."

- Line 171 : please provide the exact size that was subcloned rather than an approximation (~60 bp)

- Line 175 : the acronyms of SRF and ELK-1 are not provided.

- Figure 3f : please put the enhancer sequence in red as in figure 2, for consistency.

- Line 240 : please clarify the reporter in JET cells. Is it GFP ? If so, does this mean that JET cells were sorted based on GFP expression in Figure 4d?

- Line 272 : please correct as p19 ELISA allows to quantify particles but not to "analyze infectivity"

- Figure 5a and line 308 : please correct the figure as cells used are JET cells and not Jurkat.

- Line 316 : please clarify what do you mean by "near" ISs. How many kb around IS are you analyzing ?

- Line 593 : please put the full acronym of EMSA.

- Line 695: please specify that these are 293T cells

- Figure S3b and lines 713-715 : please explain this figure, add %.

Reviewer #3:

Remarks to the Author:

Matsuo, et al. sought to uncover the regulatory mechanisms for the HTLV-1 antisense transcript. They were able to show that there is a nucleosome free region (NFR) within the tax gene using MNase-seq along with DNA-capture-seq. Using these methods, they demonstrate that the NFR is present in both cell lines and donor samples. The authors also demonstrated enhancer function of the NFR using luciferase reporter assay showing that both sense and antisense orientation enhanced promoter activity. ChIP-seq and NET-CAGE were also performed to detect histone related modifications and enhancer RNAs near the NFR, supporting their conclusion that the NFR is an enhancer. Next, to understand how the NFR functions as an enhancer, ChIP-seq was used to show that transcription factors, SRF and ELK-1, bind to the NFR. EMSA analysis was also used to confirm that these transcription factors bind to the NFR. Finally, they found that the viral enhancer in the NFR can induce host transcription near the site of integration. RNA-seq analysis was revealed read-through transcripts around the integration sites. The authors conclude that the enhancer region they discovered likely functions to drive antisense transcription and cooperates with a viral insulator to inhibit the spread of heterochromatin from the 5'LTR. Hence, the enhancer contributes to viral persistence by upregulating HBZ transcription. They also suggest that the ability to recruit SRF and ELK-1 enables the virus to be latent but also allows reactivation to spread to an uninfected host. Finally, they point out that the ability of the viral enhancer to alter host transcription, especially near cancer-related genes, can contribute to the selection of those specific infected cells for clonal expansion during leukemogenesis. The paper is well written with no obvious flaws in the approaches and constitutes a major advance in the field. I just have a few minor comments:

Please clarify Lines 132-133 and figure 2c: "The promoter activity of the 5'LTR was enhanced but by a much smaller factor than that observed for the 3'LTR (Figure 2c)." While there was a statistically significant enhancement of the promoter at the 5'LTR when the NFR is added in the reverse orientation, there is a decrease in promoter activity when inserted in the forward orientation.

Perhaps include p-values for the following statement in Lines 133-135: "T-cell stimulation with TNF- α or Phorbol 12- Myristate 13-Acetate (PMA)/Ionomycin did not enhance promoter activity but marginally increased promoter/enhancer activity (Figure 2d)."

Lines 137-139 and figure 2e: "Chromatin immunoprecipitation sequencing (ChIP-seq) signals of enhancer-related histone modifications, including H3K27Ac, H3K4me1, and H3K4me2, were high around the NFR in ED cells (Figure 2e, upper panel)." Can you make this claim for H3K4me1?

Figure 5d and Figure 6e and f: Colors are difficult to distinguish. Maybe label major groupings to make it easier.

Lines 428-431: "Previous reports demonstrated that the HTLV-1 provirus tends to integrate near cancer-related genes in ATL cells, indicating that aberrant host genome transcription by viral integration may contribute to the multistep oncogenic process induced by HTLV-1 infection."

Saying "tends to" makes it sound like proviruses are selectively integrating near cancer-related genes which I don't think is true. Maybe should say that integrations near cancer-related genes are selected in ATL cells?

Figure S1 and Line 229-231 "The nucleotides substitutions could change stability of mRNA and translational efficiency, but we confirmed that introduction of mut1, mut2, or mut3 did not change Tax protein levels (Figure S1a)." Visually it looks like Tax might be slightly upregulated for mut1 compared to wt.

Figure 6, Figure S3 and section starting line 357. Please clarify whether 5 ATL cases (as described in text) or 8 ATL cases (as shown Fig S3a) were evaluated. When you specify ATL cases, are you describing a specific patient sample or are you describing a clone with a specific integration site (i.e. ATL-1 chrom 18: 453943)?

Matsuo et al: reply to reviewers' comments:

REVIEWER COMMENTS

Reviewer #1 (Remarks to the Author):

The study by Matsuo and colleagues describes a previously unidentified viral enhancer in viral genome of the HTLV-1, the causative agent of Adult T-cell leukemia (ATL). The authors show binding of transcription factors SRF and ELK-1 to this enhancer region in ATL-derived cell lines and freshly isolated leukemic cells. Using a cloned provirus containing mutations in the SRF- and ELK-1-binding sites they observe decreased chromatin openness at the viral enhancer, reduced viral gene transcription and changes in host gene transcription near the viral integration site in mutant virus infected cells. scRNA-seq of ATL cells revealed aberrant host genome transcription nearby integration sites of full-length and 5'LTR-deleted defective proviruses containing this enhancer.

This is an interesting study that reveals and explores a new regulatory element in the HTLV-1 proviral genome. While the work is attractive and of interest for the HTLV-1 research community, it comes with a number of questions and concerns.

Reply:

We appreciate the comment "This is an interesting study that reveals and explores a new regulatory element in the HTLV-1 proviral genome. We also appreciate the reviewer's constructive comments and suggestions.

Main comments:

Title:

What do the authors mean by "hidden"? We agree the enhancer region was unidentified until now, but is it really hidden?

Replace retroviral by HTLV-1 (enhancers in other retroviruses have been identified previously, including in ERVs)

By viral DNA capture seq approach: basically, the viral enhancer element and its main characteristics (chromatin marks, TF binding, ...) were revealed via classical methods including CHIP-seq, NET-CAGE, RNA-seq, scRNA-seq etc ... with the help of the viral DNA capture method previously developed by the authors. Viral DNA capture-seq was used in combination with these methods to target and enrich viral sequences in a subset of experiments.

Reply:

According to suggestions from reviewer #1 and #2, we created new title “A novel HTLV-1 enhancer sustains proviral expression and induces aberrant gene transcription in the integrated host genome”.

Novelty:

While the viral enhancer is novel in the context of HTLV-1, the presence of viral enhancers within retroviral genomes is not new. It was previously shown that such enhancers influence host gene expression in the vicinity of their genomic location. This is also the case for endogenous retroviral elements as demonstrated by recent work by Ozgen and colleagues (Nat Comm 2020). This work reveals that ERV-derived regulatory sequences are a source of enhancers that may be exploited by cancer cells to help drive tumour heterogeneity and evolution.

Reply:

Thank you for the thoughtful and important suggestion. We referred to the previous report that described how retroviral enhancers could contribute to oncogenic transcriptional circuits and cited the study in our revised manuscript. Our study here further demonstrated another similar phenomenon which is caused by an exogenous retrovirus HTLV-1. In addition, the intragenic enhancer identified in this study is not a typical retroviral enhancer as they are generally located in the LTR region. We have added these points in the discussion in the revised manuscript.

Line 109

We further asked if the NFR is also observed in in vivo samples in addition to the in vitro cell lines by analyzing peripheral mononuclear cells (PBMCs) freshly isolated from ATL patients and an asymptomatic carrier.

ATL patients and AC: how were they selected?

Patient primary samples and their characteristics are not described in the manuscript while several ATLs and AC samples are used in the study. ATLs are defined as “smoldering, acute, chronic” in Fig 1. In Figs 5 and 6, ATLs have numbers ranging from 1 to 8. Are some of these patient samples identical? Providing a table of patient samples used in the study with their clinical and molecular characteristics will increase clarity.

Reply:

We apologize for not providing enough information about clinical samples we used in this study.

Firstly, we added information about clinical subtype of ATL (smoldering, chronic, acute, lymphoma) to the introduction for the readers who are not familiar with ATL. Then, we have re-organized clinical sample ID and made a new table to summarize the characteristics of clinical samples, including proviral load, clinical subtype of ATL and clonality of infected cells (Supplemental Table 1).

Basically, we selected available clinical samples from our stock of viable cells without any bias.

Fig 1: degree of nucleosome freeness, how is the total input DNA-seq value computed?

Reply:

Apologies for the lack of information. We prepared fragmented DNAs by sonication and used them as input DNAs.

We have added the following text in the method.

We prepared fragmented DNAs by sonication using a Picoruptor (Diagenode s.a., Liège, Belgium) and use them as input DNAs. The degree of nucleosome freeness was calculated as the MNase-seq value normalized to the input DNA value.

Can the authors comment on the observation that in vivo patient samples (AC and ATLS) have similar MNase-seq profiles than the sense RNA-producing cell line (TBX-4B)? In both patient types, and more specifically in ATL cases, sense transcription is barely detected (this is mentioned in the introduction and repeated many times in the manuscript). Why is this not the case for NFRs? Is there a bias generated by the method? How were these samples processed?

Regarding the insulator region, the NFR at this location is not significantly more represented than other regions all over the provirus, except for ED. Same question as above.

Reply:

Re: similarity of MNase-seq profiles between in vivo samples and TBX-4B.

HTLV-1 provirus DNA in ED is heavily methylated¹, so MNase cannot access the 5' side of provirus, resulting in low MNase-seq signal. A portion of fresh ATL cells and cell lines with high level of sense transcription exhibited relatively open chromatin structure^{1,2}. That made the difference of MNase-seq signals between ED and fresh ATL or TBX-4B cells. We have added more explanation in the revised text (line 117-120).

Re: In both patient types, and more specifically in ATL cases, sense transcription is barely detected (this is mentioned in the introduction and repeated many times in the manuscript). Why is this not the case for NFRs?

Sense proviral expression in ED is irreversibly silenced by DNA methylation. However, HTLV-1 provirus in fresh ATL cells can be reactivated by ex vivo cultivation³. These findings suggest that proviral silencing mechanism in fresh ATL cells is different from ED. Silencing of the sense strand expression in fresh ATL cells is possibly not due to nucleosome status but by other unknown mechanism. However, regardless of different nucleosome status of 5'LTR, the 3'LTR is open and antisense transcription is active in ED, TBX-4B and fresh ATL cells, which is consistent with the presence of the NFR in these cells.

Re: Is there a bias generated by the method? How were these samples processed?

Fresh PBMCs were more susceptible to MNase treatment than cell lines. Thus, we used different protocols for cell lines and fresh PBMCs as described in Method. We confirmed that the performance of MNase treatment was similar between cell lines

and fresh PBMCs by gel electrophoresis after MNase digestion. We have added the information in the Method.

In IGV figures, explain the numbers between brackets (scale, coverage) in figure legends. Readers not familiar with IGV representations may get lost.

Reply:

Thank you for valuable suggestions. We have added the following text in figure legend. The range in square brackets indicates the range of read counts.

CHIP-seq and Fig 2: can the authors comment on the resolution of their CHIP-seq experiments? The CHIP-seq data and images are not fully convincing. There is a broad coverage of Chip-seq reads. How was the specific attribution to the NFR defined? It seems that the reads are covering the entire HBZ transcript region, or did I miss anything?

Reply:

We showed the CHIP-seq result on the IGV by zooming in on the HTLV-1 provirus, which has only about 9kb length. That makes the pattern a little bit atypical. To solve this issue, we have revised the figure by showing both the zoomed in and zoomed out version (Fig. 2e-f and Supplemental Fig. 2a-c). We believe that peaks in the revised figures are clear enough for the readers.

Chromatin immunoprecipitation sequencing (CHIP-seq) signals of enhancer-related histone modifications, including H3K27Ac, H3K4me1, and H3K4me2, were high around the NFR in ED cells (Figure 2e, upper panel).

What is meant by around?

Reply:

We corrected this from “around” to “in the proximity of” in the revised manuscript.

Luciferase experiments: how does luciferase activity change when Tax expression is added? The in vivo data in Fig 1 (ATL, AC) show sense-related NFR and possibly some

positive strand expression, how would tax influence promoter activity with or without the enhancer region on both the 5' and 3' promoter (LTR)? Would this better reflect the situation in vivo, where despite a low sense activity, CTL responses to tax reveal a persistent tax expression.

Reply:

We performed the promoter/enhancer assay with and without Tax. The result showed enhancer increased the 5' and 3' promoter activity with Tax expression, although the effect is subtle. We show the new data as Supplemental Fig. 1b.

The legend to Fig 2 is unclear: what do the authors mean by the nucleosome region? NFR? Nucleosome-free?

Fig. 2 The nucleosome region harbors enhancer-related histone modifications and produces enhancer RNAs.

Reply:

We corrected this. The new title is "The nucleosome-free region harbors enhancer-related histone modifications and produces enhancer RNAs."

Fig 2e: the NET-CAGE experiment generates interesting results, however the data could be commented in greater detail. It would also be useful to see this method applied to TBX-4B. Why was it not performed? Again, in vivo, we also expect finding proviruses that are not impaired in sense transcription by mutation or epigenetic silencing, not only "ED-like" proviruses. Performing NET-CAGE with patient samples (this was also done in the case of sensitive CHIP-seq, below) would be an asset and biologically relevant.

Reply:

Thank you for valuable suggestions and comment that the NET-CAGE experiment generates interesting results. We performed NET-CAGE experiment using TBX-4B cells. There was much more CAGE signal in the TBX-4B than in ED cells, because of the high levels of sense transcription in TBX-4B in comparison to ED cells (Fig. 1a). As we

previously reported, this is also explained by the susceptibility of HTLV-1 RNA towards ZAP-mediated RNA processing⁴.

According to reviewer's suggestions we performed NET-CAGE analysis by using clinical samples. However, the data quality of the results we obtained was very poor because of limitation of sample availability for NET-CAGE, (data not shown).

Line 171

The NFR region we identified in this study is ~160 bp in length

How this was this determined?

Reply:

Before we did MNase-seq, we performed qPCR to analyze nucleosome-free region. As shown below, we observed remarkable reduction of NFR signal from positions 8,000 to 8,160 in the HTLV-1 provirus of ED cells. Based on this finding, we designated the nucleotides from positions 8,000 to 8,160 (160 bp) as the NFR in this study.

Fig 3:

a The prediction of transcription factor binding to the NFR was performed by using TFBIND (<http://tfbind.hgc.jp/>). Candidate transcription factor binding sites are shown.

Not clear what the numbers 8000 and 8160 represent? How do they relate to the indication of 7100 earlier in the manuscript?

Reply:

We apologize for unclear description in the initial manuscript. As we described above, the numbers 8000 and 8160 represent the position in HTLV-1 provirus. We have added that in the figure legend.

The number 7100 is the position of HTLV-1 insulator, which is reported previously⁵.

Why were SRF and ELK1 chosen among all candidates? Were they tested in HTLV-1 viral capture CHIP-seq described by the authors as being very sensitive?

Reply:

We apologize for not explaining the details in the initial manuscript. We tested other candidate factors, including CREB, MEF2, GATA-3, STAT-3 and ATF-4, by ChIP-qPCR. STAT-3 showed some signal in the NFR. We next performed STAT3 ChIP-seq to investigate if STAT3 binds to HTLV-1. The result showed a broad and weak signal in HTLV-1 provirus, but the peak was not statistically evident. As we show below, there were clear, sharp peaks in the positive control region with statistically significant but not in HTLV-1 provirus, indicating that STAT3 does not bind to HTLV-1 provirus. We also analyze other factors but there were no clear evidence that they bind to the NFR.

We added this description in the revised manuscript.

How was this performed? If I am correct, the CHIP-seq capture-seq combination method is not described.

Reply:

We apologize for unclear description in the initial manuscript. We added the method in the revised manuscript.

Line 225

(The) SRF and ELK-1 plays a critical role in HTLV-1 enhancer function

Fig S1a Tax protein levels in nuclear lysates of cells transfected with wt or enhancer region-mutated Tax-expression vectors.

Was this tested with expression vectors, or using full-length proviruses? How was this experiment performed? Is this described in Methods?

Reply:

We apologize for unclear description in the initial manuscript. We added the details to the method in the revised manuscript.

Suppl Fig 1a. Where is SFig 1b and potentially c, d, ...?

Reply:

We have modified this point in revised manuscript.

After quantifying viral gene expression in the transiently transfected cells and viral production in the culture supernatant (Figure 4a), we found a marginal decrease of p19 production in the supernatant of mut plasmid-transfected cells; however, there was no statistically significant difference (Figure 4b). Nevertheless, there was a significant reduction in tax and HBZ expression at the mRNA level (Figure 4c).

Tax expression has an impact on 5' LTR promoter activity. The authors observed decreased tax expression, but no significant change in p19 expression. Could they comment on this?

Reply:

As we described in the revised manuscript, we would like to propose the following two effects of the enhancer on the proviral expression.

- (1) the enhancer affects antisense transcription by functionally interacting with the HTLV-1 3'LTR promoter
- (2) the enhancer also affects both sense and antisense transcription by changing the chromatin openness of HTLV-1 provirus.

These should be the reason why there is a difference between wt-NFR and mut-NFR in some assay but not in other assays.

In figure 4b and 4c, we measured p19 production and mRNA expression in 293T cells transfected with wt or mut HTLV-1 plasmid. In this assay, the HTLV-1 template DNA was not integrated but present as an unintegrated plasmid. As such, the effect of NFR mutation on sense transcription (Tax) was not so evident when compared with the cases containing an integrated HTLV-1 provirus, such as the result in Figure 4i.

In Figure 4c, we extracted RNA from 293T cell transfected with 1ug HTLV-1 plasmid DNA (wt or mut) 1 day after transfection. There was a similar tendency – NFR-mutation decreased p19 and tax transcripts – but there was a difference in statistical analysis. In general, the change of mRNA levels are faster than that of protein levels. Thus, the statistical difference between figure 4b and 4c could be possibly explained by the different assay systems used.

JET cell infection experiment: if tax is decreased in mutant proviruses, why is it not the case in infected cells?

Reply:

I may miss the reviewer's point, but there was decrease of tax in mutant-infected cells (Figure 4i).

Line 245

The proviral load of HTLV-1-mut-transfected JET cells was lower than that of HTLV-1-wt-transfected ones (Figure 4f).

It is not clear if the authors produced both transfected and infected JET cells. Is 'transfected' correct, or are these cells infected? Please clarify.

Reply:

We apologize for unclear description in the initial manuscript. We only produced infected JET cells. We have changed the text from 'transfected JET cells' to 'infected JET cells'.

How do the authors explain reduced PVL and is this important? PVLs in Fig 4 suggest multiple IS per cell (>2) for wt. Since the authors examine bulk data, how could they determine the distribution across clones? Cells are sorted according to the integrated reporter plasmid (tax response element and fluorescence) activity, suggesting they rely again on tax expression that is said decreased in mut proviruses.

Reply:

Thank you for critical comment. We do agree with the point the reviewer suggested. There is a possibility that the PVL was biased by sorting step. We added the possibility in the revised manuscript.

Distribution of viral IS was not so different between the WT and mutant HTLV-1-infected JET cells in terms of the relationship with the host gene and epigenetic environment (Figure S2a and S2b).

Wouldn't the opposite be expected if the impact of NFR on neighboring genomic regions provides an advantage to the cell? If the presence of the viral enhancer is critical, we may expect seeing the selection of cells that have increased fitness, even *in vitro*?

Reply:

This is result of transient experiment *in vitro*. Thus, the viral IS distribution depends mainly on integration preference rather than selection of infected clones, where the HTLV-1 enhancer plays a role.

Fig 4 k MNase assay of in JET cells infected with HTLV-1-wt (above) or mut (below).

Fig 5: in JET clones wt and mut

Line 316

The proportion of upregulated genes in JET clones infected with HTLV-1-wt was significantly higher than those in mutant HTLV-1 clones ($P < 0.01$; Figure 5d).

Is this related to overall expression or is this due to transcripts in the neighborhood of HTLV-1 ISs?

Reply:

Expression level of each gene was analyzed by RNA-seq. The value is generated as TPM (transcript per million reads). To calculate TPM, we normalized for gene length first,

and then normalize for sequencing depth second. Therefore, the upregulation observed is not due to overall expression.

Line 321

We then used CRISPR/Cas9 to introduce the mutation that abrogated SRF-ELK-1 binding to the enhancer region (Figure 3d and 3e, Figure 4g) of a clone infected with wt-HTLV-1. SRF/ELK-1 ChIP-seq peaks in wt-HTLV-1-infected cells were abolished in the CRISPR-mutated cells, thereby reducing proviral transcription both at sense and antisense direction (Figure 5e and 5f) and chromatin openness in the enhancer region (Figure 5g).

How many IS were found in this clone and have all (if multiple) corresponding proviruses been edited by Crispr/Cas9?

Reply:

We used wt_5 clone in this analysis. As shown in Supplemental Table 2, the clone has one provirus.

Fig 5: limiting dilution

Does limiting dilution introduce bias? Fit clones will have a significantly higher chance to survive the cloning procedure.

Was there only one clone selected for wt and one clone for the mut condition? How many IS? Previous data suggest the number of IS per clone is > 1 . Did the authors in purpose select a clone with only 1 IS?

Reply:

Thank you for critical and constructive comments.

We cannot exclude the possibility than some bias was introduced during the cloning step. Thus, we analyzed additional clones and presented the result as new data (see next reply). We showed the characteristics of each clones in Supplemental Table 2. Data of wt_5 clone was shown in Fig. 5b. Data of mut_1 clone was shown in Fig. 5c.

The mut provirus Jurkat clone has an IS in a gene that is expressed in parental cells (mRNA, RNA-seq) while the wt provirus has an IS in a gene that does not show expression (parental RNA-seq). Is it correct to compare these two cases? We suggest examining several clones with ISs located in both expressed and non-expressed genomic regions for both situations (wt and mut). In summary, this is a very interesting experiment, however, it is statistically unsound. A larger number of clones should be examined, not just one nice example of each condition (wt and mut). This would significantly strengthen the authors' conclusions.

Crispr/Cas mut clone: which wt clone was used for this experiment? Same clone as the one illustrated above (Fig. 5)?

Reply:

Thank you for critical and constructive comments. We do agree with the reviewer's point. Proviral and host genome expression is controlled not only by proviral sequence but also by integrated host genome. It is not correct to compare wt and mut in different clones, because their integration sites are different. Thus, we performed Crispr/Cas9 experiment to compare wt and mut HTLV-1 in same integration site. To confirm reproducibility, we generated additional clones having provirus in the same integration site as original clone but the NFR contains mutation. The result demonstrated that proviral expression was remarkably suppressed in the mutant clone. We showed the result as new data (Supplemental Fig. 5).

Primary cell studies:

Line 359

We further investigated the effect of HTLV-1 integration on viral and host genomes by performing mRNA-seq analysis using freshly isolated PBMCs from five ATL cases.

How were ATL cases selected, and how do they relate, if they do, to the primary samples shown in Fig. 1 (see comments above)?

Reply:

We selected available clinical samples from our stock of viable cells without any bias. The characteristics of clinical samples are shown in Supplemental Table1. ATL1 and ATL2 were analyzed by MNase-seq (Figure 1d), ChIP-seq (Figure 3b), and single cell RNA-seq (Figure 6b-c). Not all assays are used to analyze the ATL cases due to limited cell numbers.

Line 367

More importantly, there were clear peaks of SRF and ELK-1 ChIP-seq signals in integrated proviruses, indicating SRF and ELK-1 play a role in the transcriptional regulation (Figure 6b–6d).

Was this explored by what the authors called sensitive Capture-mediated CHIP-seq?

Reply:

The SRF and ELK-1 ChIP-seq were performed with and without HTLV-1 DNA-capture (Figure 3b and Figure 6b-c, respectively). We added this information in the figure legend.

Line 380

Furthermore, there was a significant increase of the local transcriptome in the ATL clone but not in non-ATL CD4+ T cell clones (Figure 6g and 6h, Figure S4a-4c, right panels).

How many single cells were examined for each ATL?

Reply:

ATL-1: 476 cells,

ATL-8: 7452 cells.

We add this information in revised figure.

In Fig 6 g and h (scRNA-seq data in IGV), does each IGV graph represent a single cell?

Reply:

Data shown in figure 6g-h was cumulative data obtained by single cell analysis. We have added the number of cells used in each analysis in the revised figure.

How many did the authors screen? Are none of the genes in the vicinity of the IS in ATL 1 or ATL 6 expressed in uninfected T-cells or HTLV-1 infected non ATL clones (COLEC12, MRPL13, MTBP)? Same comment, how many non ATL or uninfected T-cells did the authors screen?

Reply:

We think this should be the comments for Fig. 6g-h. IGV image is not one single cell data but cumulative data. We apologize for the confusion. To avoid that, we presented number of cells used for IGV image in revised figure.

What is the percentage of single cells belonging to the ATL clone (defined by matched TCR) that showed a transcriptome profile identical to that illustrated in Fig 6 in the IS vicinity?

Reply:

ATL-1: 70.5% in CD4+ T cells

ATL-8: 98.4% in CD4+ T cells

As mentioned above, the IGV images are not one single cell data but cumulative data. Since single data is sparse, we cannot evaluate transcriptome profile at each single cell.

Is it expected that all ATL cells express these local transcripts at a given time?

Reply:

Since single data is sparse, it is impossible to discuss about this point.

Did the authors detect HBZ transcription in all single cells? If not, in what percentage of cells and how does the level of HBZ transcript expression compare to the distribution of host cell transcripts?

Reply:

We analyzed ATL8 data from the reviewer's point of view. Since single data is sparse, it is not easy to discuss about this point.

We opened our single cell study as a preprint paper

(<https://biorxiv.org/cgi/content/short/2021.09.01.458291v1>). Please see that for more in details.

It should be possible to examine the local transcriptome in single cells belonging to other smaller clones since the authors claim that the viral capture-seq method can identify the majority of IS including the non ATL integration sites.

Reply:

This is very interesting and important point.

In single cell analysis, we did not use IS but TCR information to identify clones. HTLV-1 DNA-capture-seq was performed using bulk DNAs from fresh PBMCs from individuals. We can obtain clonality of infected cells (Supplemental Fig. 6). The data is bulk information not single cell resolution; therefore, this is not feasible analysis.

Do the authors find transcriptional changes in the vicinity of these ISs by scRNA-seq?

Reply:

This is exactly what we showed in Figure 6g-h.

But please let me emphasize again that the graph is cumulative value from multiple cells. We have added the cell number used in each analysis in revised figure.

Is modification of the transcriptome around the IS a specific feature of the transformed ATL clone? Is it related to tumor progression?

Reply:

This is extremely interesting and important point, but the data we showed in this study is not enough to answer. We would like to investigate these issues as next study.

Generally speaking, the in vivo experiments described in the last section of the manuscript are of substantial interest, but the experiments are poorly documented especially the scRNA-seq study. Both the experimental design (including information about the number of cells examined) and the bioinformatics analysis (i.e. analysis of the spliced transcripts in the vicinity of the IS in single cell data, provirus-driven transcript modifications) require more detailed explanation to improve clarity.

Reply:

We do apologize for not providing enough explanation. We have corrected these issues as much as we can in the revised manuscript.

Do the authors expect being able to identify all or the majority of transcripts that have a junction between the virus and the host genome using 10X Genomics scRNA-seq? Is the standard configuration of 10X Genomics (91-98 bp reads) appropriate for the detection of these transcripts at the single cell level?

Reply:

Since single cell gene expression data is sparse, we rarely detected transcript containing a junction between the virus and the host genome. Instead, we used TCR information to identify ATL clone in this study. To avoid confusion of readers, we have added the details in the method as below.

‘In the single cell analysis, we defined ATL cell as the most expanded CD4⁺ T cell clone in the sample based on TCR information.’

Is the viral enhancer capable of driving host gene transcription on its own, thus in combination with host gene promoter(s) and regulatory regions but without the

implication of the viral promoter, or does it require the HBZ promoter to exert its function?

Reply:

Thank you for constructive and critical suggestions. To address these points, we performed additional promoter/enhancer assay with viral insulator region. The construct with HBZ promoter-enhancer-insulator in the same orientation as the provirus exhibited the highest activity, indicating that a combination of these three factors is optimal for the enhancer activity. In addition, it has been reported that the 5'LTR is frequently deleted but not 3'LTR. That indicates that these viral regulatory elements, including insulator, enhancer, and 3'LTR promoter plays indispensable role in maintaining proviral expression.

The impact of defective HTLV-1 proviruses on the local host transcriptome in ATLs carrying a 5'LTR deleted provirus has been previously reported (Kataoka, 2015, Rosewick, 2017).

Reply:

Thank you for valuable suggestion. We have mentioned and added previous studies as reference papers.

Minors comments:

Several errors in Methods, Figure legends, Discussion. Examples in Fig 1: Stranded proviral transcriptome (s?) are; in Methods: ED, 293T, Jurkat, and JET cells infected (with) WT or mutant HTLV-1 molecular clones; in Discussion: Consistent with this notion, the intragenic viral enhancer and insulator are maintained even in defective type proviruses that is (are) observed (in) 20-30% of ATL cells; thereby upregulates cancer-related genes near ISs and might contributes (contribute) to the selection of a specific infected cell for clonal expansion

Reply:

We apologize for the errors. We checked and corrected all the points raised by the reviewer.

Reviewer #2 (Remarks to the Author):

The manuscript by Matsuo et al. aimed at identifying nucleosome-free DNA regions (NFR) in the HTLV-1 provirus sequence and investigating their impact on HTLV transcriptional activity, both in the sense and antisense directions. Using micrococcal nuclease digestion coupled with enrichment of HTLV DNA sequences, they identified such a NFR region at the 3' end of the proviral sequence, between an insulator sequence and the 3'LTR promoter element. They showed that this NFR sequence was indeed able to enhance the 3' LTR-mediated transcriptional activity, but not the 5' LTR one. They further identified SRF and ELF-1 as transcription factors binding to that NFR and regulating its activity (via CHIP-Seq analyses, EMSA and mutagenesis experiments). Finally, they provided transcriptomic data showing the impact of HTLV provirus insertion, complete or defective, on transcription activity in integration site surrounding region (readthrough).

Overall, the study provides a plethora of experiments confirming the identification and relevance of this novel enhancer element. However, a few issues need to be answered to ensure the absence of bias in the analysis and the pertinence of the conclusions. Furthermore, the manuscript might be improved with additional information to facilitate reading and understanding.

Major issues :

- One of the aim is to identify a putative role of the NFR enhancer on the sense and antisense transcription. Although there is no doubt that this NFR is playing a role for antisense transcription, NFR impact on sense transcription seems minimal in the luciferase assay (Figure 2c). However, I am wondering if in this assay the design could be biased as the enhancer is used with the complement 3'U5 region only vs the 5' whole LTR on the other hand.

Furthermore, it may be useful to include in Figure 2c constructs that also include the insulator before and after the enhancer as controls, to further mimic the proviral sequence.

Reply:

Thank you for constructive comments and suggestions. We additionally performed promoter assay using 5' whole LTR and 5'U3 region to evaluate the effect of the NFR on sense transcription from HTLV-1 provirus. The effect was not different between 5' whole LTR and 5'U3 region. We showed this result as new figure (Supplemental Fig. 1a). Also, we introduced insulator region in the reporter construct and performed promoter assay and found that insertion of insulator region increased promoter/enhancer activity in both sense and antisense transcription. We showed the result as new figure (Supplemental Fig. 1b).

On another hand, could you please clarify the impact of this NFR on the sense transcription as results from figure S1 and figure 4i are confusing, and potentially contradictory.

Reply:

Thank you for critical comment.

RE: figure S1 (Extended Data Fig. b in revised figure)

What we would like to show in Supplemental Fig. 3b is the effect of the introduced mutations on the Tax protein expression levels as the NFR is present within the coding sequence of the Tax protein. Here, our results showed that there was no change observed in Tax protein levels upon NFR mutation (Supplemental Fig. a-b).

RE: figure 4i

We showed a remarkable reduction of Tax expression in JET cells infected with mutant HTLV-1 when compared with JET cells infected with wt HTLV-1 (Figure 4i).

We think that there are two independent actions of the enhancer on proviral transcription as followings,

- (1) the enhancer affects antisense transcription by functionally interacting with the HTLV-1 3'LTR promoter
- (2) the enhancer also affects both sense and antisense transcription by changing the chromatin openness of HTLV-1 provirus integrated in the host genome.

Possibly because of reason (2), we observed the remarkable reduction of both tax and HBZ expression (Figure 4i).

- The data obtained from ChIP-Seq for histone modification marks (Figure 2e) are not very conclusive and the authors should be careful interpreting these data (lines 137-139). Indeed “high” levels of histone marks “around” the NFR cannot be stated in my opinion

.

Reply:

We showed the ChIP-seq result on the IGV by zooming in HTLV-1 provirus, which has only about 9kb length. That makes the pattern a little bit atypical. To solve this issue, we have made revised figures by showing both zoomed in and zoomed out version (Fig. 2e-f and Supplemental Fig. 2a-c). We believe that peaks in the revised figures are clear enough for the readers.

- Analysis of the NFR region provided multiple putative transcription factors binding to it. It is not clear why the authors decided to investigate only SRF and ELK-1. Indeed, the other candidates should be investigated as well as otherwise the whole study can be misleading. Although there is no doubt that SRF and ELK-1 bind to this NRF enhancer, there might not be the instigators of the transcription seen. Indeed, it could well be that Stat-3 for instance is also binding and that the mutations generated also prevent its binding and activity.

Reply:

We apologize for not explaining the details in the initial manuscript. We tested other candidate factors, including CREB, MEF2, GATA-3, STAT-3 and ATF-4, by ChIP-qPCR. STAT-3 showed some signal in the NFR. We next performed STAT3 ChIP-seq to investigate if STAT3 binds to HTLV-1. As we show below, there were broad and weak signals in HTLV-1 provirus, but there were clear, statistically significant peaks in the positive control region, such as the promoter region of *IRF4* and *FOS* genes, indicating that STAT3 does not bind to HTLV-1 provirus. We also analyze other factors but there were no clear evidence that they bind to the NFR. We added this description in the revised manuscript.

- In my opinion, the authors speculate on the function of the NFR enhancer on sense/antisense transcription and cooperation with CTCF (lines 416-419 and 434-435), without having been able to completely and formally demonstrate it,

Reply:

We agree with the reviewer about the point.

RE: lines 416-419

To see the relationship between insulator and promoter/enhancer, we also analyzed the promoter/enhancer activity with or without the viral CTCF insulator⁵. Insulator increased the promoter/enhancer activity (Fig. 2c).

RE: lines 434-435

Melamed A et al demonstrated that HTLV-1 alters the structure and transcription of host chromatin in cis possibly via CTCF-mediated chromatin looping. In current study, we propose that the enhancer in the HTLV-1 provirus plays a role in HTLV-1 mediated upregulation of the host genome. We have modified the sentence so as not to cause any misleading impression.

- Title suggestion: Identification and characterization of a novel HTLV-1 enhancer by a viral DNA-Capture-Seq approach

Reply:

Thank you for valuable suggestion. I agree that the title is important for the paper. According to suggestions from reviewer #1 and #2, we changed the title to “A novel HTLV-1 enhancer sustains proviral expression and induces aberrant gene transcription in the integrated host genome”.

- The introduction should provide more background information, especially between the different forms of ATL (acute, chronic, smoldering).

Reply:

We modified introduction according to suggestions from the reviewer.

- The methods are very superficial and do absolutely not allow to reproduce the experiments presented as such. Cell, plasmid, or compound concentrations are missing as well as durations or volumes; for instance how much MNase was added, antibody dilution at 1:250, transfection conditions, amount of plasmid transfected, puromycin concentration and days of selection, etc. A section about plasmids and their description might be introduced. For instance, when performing ChIP-Seq experiments, is DNA-capture performed after or before the ChIP ? When assigning reads to the provirus sequence, how did you handle the assignment to 5' vs 3'LTR regions?

Reply:

We do apologize for not providing necessary information. We have carefully reorganized and added more information to Method.

- Figure legends are incomplete and often do not contain all the descriptive information. For instance, in the IGV viewers, please specify what are the numbers in square brackets. Figure 2f, what is the color coding, and the upper and lower graphs in sense and then antisense transcripts?

Reply:

We apologize for not providing necessary information. We have checked again and added more information in Figure legends.

- Figure 1c : please add a scale with nucleotide position on top of the provirus sequence, indicate the start and end of each region (LTR; insulator, NFR).

Reply:

We have modified the figure according to suggestions from the reviewer.

- Lines 133-135 about Figure 2d : please stick to the facts as here data are not significant, so the text cannot state that there is a marginal increase.

Reply:

We have changed the text according to suggestions from the reviewer.

- Figure 3d: please provide the complete and exact sequence as supplementary information.

Reply:

We have provided the complete and exact sequence in Supplemental Fig. 3a.

- Figure 4h : please provide a supplementary table with the list of integration sites and features.

Reply:

We created a supplementary information with the list of integration sites regarding Figure4h.

- The rationale for analyzing CTCF (lines 318-320 and figure 5d) is not clear, please provide more information.

Reply:

We have changed the text to show rationale for analyzing CTCF.

- ATL cells from patients (Figure S3) : please provide additional information regarding the ATL degree (acute, smoldering, ...)

Reply:

Apologizes for insufficient information. We have created new table to explain the details of ATL cases (Supplemental Table 1).

- Figure 6a : please include all patients, why only ATL-1 to 5 ?

Reply:

This is the data from bulk mRNA-seq data from ATL cases. Due to limitations on sample availability, we only show here the results for the ATL cases that we are able to analyze. We added the details in the figure legend.

- Figure 6e : please explain in the methods which data did you use to annotate PBMC subset.

Reply:

Clusters were annotated based on expression level of known marker genes such as CD3D for T cells and CD79A for B cells. We add the details in the revised Method.

Minor issues :

- Lines 33-35 : the sentence is confusing, as it indicates that antisense transcription (thus from the 3'LTR) can be active event if the 5'LTR sequence is defective. I don't understand this link as antisense transcription should not be dependent from a functional sense transcription to my knowledge.

Reply:

5'LTR is a known enhancer/promoter and the promoter of *tax*. What we would like to mention here is that anti-sense transcription is maintained even in the absence of the known enhancer/promoter and *Tax*, indicating that known 5'LTR promoter/enhancer might not play indispensable role in antisense transcription. Thus, there should be other mechanisms to maintain antisense transcription.

- Line 113 : please rephrase "... in naturally virus-infected cells from in individuals).

Reply:

We have changed the sentence.

- Figure 2e : please add as well the orange shadow for the picture.

Reply:

We have modified the Figure 2e.

- Line 152 : please modify “The nucleosome-free region ...”

Reply:

We have modified the figure legend.

- Line 171 : please provide the exact size that was subcloned rather than an approximation (~60 bp)

Reply:

We have modified sentence according to the reviewer’s suggestion.

- Line 175 : the acronyms of SRF and ELK-1 are not provided.

Reply:

We have provided acronyms of SRF and ELK-1 according to the reviewer’s suggestion.

- Figure 3f : please put the enhancer sequence in red as in figure 2, for consistency.

Reply:

We have modified the figure.

- Line 240 : please clarify the reporter in JET cells. Is it GFP ? If so, does this mean that JET cells were sorted based on GFP expression in Figure 4d?

Reply:

JET cell is Jurkat cell expressing tdTomato under the control of 5 times tandem repeat of Tax responsive element (TRE). We have added the information to figure legend.

- Line 272 : please correct as p19 ELISA allows to quantify particles but not to “analyze infectivity”

Reply:

We have modified the figure legend.

- Figure 5a and line 308 : please correct the figure as cells used are JET cells and not Jurkat.

Reply:

We have modified the figure according to the reviewer's suggestion.

- Line 316 : please clarify what do you mean by "near" ISs. How many kb around IS are you analyzing?

Reply:

We have modified sentence as following.

We analyzed host genes near HTLV-1 provirus, within 100 kb from viral IS and found that proportion of upregulated genes in JET clones infected with HTLV-1-wt was significantly higher than those in mutant HTLV-1 clones ($P < 0.01$; Figure 5d).

- Line 593 : please put the full acronym of EMSA.

Reply:

We have corrected this point.

- Line 695: please specify that these are 293T cells

Reply:

We have corrected this point.

- Figure S3b and lines 713-715 : please explain this figure, add %.

Reply:

We have corrected this point.

Reviewer #3 (Remarks to the Author):

Matsuo, et al. sought to uncover the regulatory mechanisms for the HTLV-1 antisense transcript. They were able to show that there is a nucleosome free region (NFR) within the tax gene using MNase-seq along with DNA-capture-seq. Using these methods, they demonstrate that the NFR is present in both cell lines and donor samples. The authors also demonstrated enhancer function of the NFR using luciferase reporter assay

showing that both sense and antisense orientation enhanced promoter activity. ChIP-seq and NET-CAGE were also performed to detect histone related modifications and enhancer RNAs near the NFR, supporting their conclusion that the NFR is an enhancer. Next, to understand how the NFR functions as an enhancer, ChIP-seq was used to show that transcription factors, SRF and ELK-1, bind to the NFR. EMSA analysis was also used to confirm that these transcription factors bind to the NFR. Finally, they found that the viral enhancer in the NFR can induce host transcription near the site of integration. RNA-seq analysis was revealed read-through transcripts around the integration sites. The authors conclude that the enhancer region they discovered likely functions to drive antisense transcription and cooperates with a viral insulator to inhibit the spread of heterochromatin from the 5'LTR. Hence, the enhancer contributes to viral persistence by upregulating HBZ transcription. They also suggest that the ability to recruit SRF and ELK-1 enables the virus to be latent but also allows reactivation to spread to an uninfected host. Finally, they point out that the ability of the viral enhancer to alter host transcription, especially near cancer-related genes, can contribute to the selection of those specific infected cells for clonal expansion during leukemogenesis. The paper is well written with no obvious flaws in the approaches and constitutes a major advance in the field. I just have a few minor comments:

Please clarify Lines 132-133 and figure 2c: "The promoter activity of the 5'LTR was enhanced but by a much smaller factor than that observed for the 3'LTR (Figure 2c)." While there was a statistically significant enhancement of the promoter at the 5'LTR when the NFR is added in the reverse orientation, there is a decrease in promoter activity when inserted in the forward orientation.

Reply:

We revised Fig. 2c according to reviewers' comment. We have changed text according to the new figure.

Perhaps include p-values for the following statement in Lines 133-135: "T-cell stimulation with TNF- α or Phorbol 12- Myristate 13-Acetate (PMA)/Ionomycin did not

enhance promoter activity but marginally increased promoter/enhancer activity (Figure 2d).”

Reply:

According to suggestions from reviewer #1 and #3 We modified the text as followings: “T-cell stimulation with TNF- α or Phorbol 12- Myristate 13-Acetate (PMA)/Ionomycin did not enhance promoter activity (P = 0.157 and 0.155, respectively, Figure 2d).”

Lines 137-139 and figure 2e: “Chromatin immunoprecipitation sequencing (ChIP-seq) signals of enhancer-related histone modifications, including H3K27Ac, H3K4me1, and H3K4me2, were high around the NFR in ED cells (Figure 2e, upper panel).” Can you make this claim for H3K4me1?

Reply:

We showed the ChIP-seq result on the IGV by zooming in HTLV-1 provirus, which has only about 9kb length. That makes the pattern a little bit atypical. To solve this issue, we have made revised figure by showing both zoomed in and zoomed out version (Figure 2e and Supplementary figure xx). We believe that peaks in the revised figures are clear enough for the readers.

Figure 5d and Figure 6e and f: Colors are difficult to distinguish. Maybe label major groupings to make it easier.

Reply:

We have modified the figures according to reviewer’s suggestion.

Lines 428-431: “Previous reports demonstrated that the HTLV-1 provirus tends to integrate near cancer-related genes in ATL cells, indicating that aberrant host genome transcription by viral integration may contribute to the multistep oncogenic process induced by HTLV-1 infection.”

Saying “tends to” makes it sound like proviruses are selectively integrating near cancer-related genes which I don’t think is true. Maybe should say that integrations near cancer-related genes are selected in ATL cells?

Reply:

We totally agree with the reviewer’s point. We have changed the sentence according to reviewer’s suggestion.

Figure S1 and Line 229-231 “The nucleotides substitutions could change stability of mRNA and translational efficiency, but we confirmed that introduction of mut1, mut2, or mut3 did not change Tax protein levels (Figure S1a).” Visually it looks like Tax might be slightly upregulated for mut1 compared to wt.

Reply:

We repeated WB analysis to make this point clear. We obtained clearer result than before and showed the result in revised paper.

Figure 6, Figure S3 and section starting line 357. Please clarify whether 5 ATL cases (as described in text) or 8 ATL cases (as shown Fig S3a) were evaluated. When you specify ATL cases, are you describing a specific patient sample or are you describing a clone with a specific integration site (i.e. ATL-1 chrom 18: 453943)?

Reply:

We apologize for that the numbering of ATL cases was not clear. We have re-organized numbering of ATL cases. To make this point clear, we have made a table summarizing characteristics of each ATL cases (Supplemental Table 1).

- 1 Takeda, S. *et al.* Genetic and epigenetic inactivation of tax gene in adult T-cell leukemia cells. *Int J Cancer* **109**, 559-567, doi:10.1002/ijc.20007 (2004).
- 2 Taniguchi, Y. *et al.* Silencing of human T-cell leukemia virus type I gene transcription by epigenetic mechanisms. *Retrovirology* **2**, 64, doi:10.1186/1742-4690-2-64 (2005).
- 3 Hanon, E. *et al.* Abundant tax protein expression in CD4+ T cells infected with human T-cell lymphotropic virus type I (HTLV-I) is prevented by cytotoxic T lymphocytes. *Blood* **95**,

- 1386-1392 (2000).
- 4 Miyazato, P. *et al.* HTLV-1 contains a high CG dinucleotide content and is susceptible to the host antiviral protein ZAP. *Retrovirology* **16**, 38, doi:10.1186/s12977-019-0500-3 (2019).
 - 5 Satou, Y. *et al.* The retrovirus HTLV-1 inserts an ectopic CTCF-binding site into the human genome. *Proc Natl Acad Sci U S A* **113**, 3054-3059, doi:10.1073/pnas.1423199113 (2016).

REVIEWER COMMENTS, second round

Reviewer #1 (Remarks to the Author):

Overall, the authors have substantially improved the manuscript and the extra experiments strengthen the conclusions. However, the changes in the revised manuscript are not highlighted, as a result, it was complicated to navigate between the rebuttal and the revised manuscript and catch all changes mentioned by the authors. While the work has been significantly improved, it still comes with some over-interpretation and concerns described below.

Main comments

Title:

"integrated genome"

The viral genome is integrated in the host genome, the host genome carries an integrated provirus, but the host genome is not "integrated"

I would suggest using a title close to the suggestion of reviewer 2, but leaving out "by a viral DNA seq capture approach" for the reasons explained in the first round of reviews. Does the enhancer have a direct effect on host transcription or is this via viral gene expression (see comments below)?

"Identification and characterization of a novel enhancer in the HTLV-1 proviral genome"

Comments regarding some of the answers to our questions/comments:

What is meant by around?

Reply:

We corrected this from "around" to "in the proximity of" in the revised manuscript
Can the authors provide a more quantitative explanation? How distant is "proximity"?

NET-CAGE data

In their reply, the authors mention they performed NET-CAGE on the TBX-4B cells, but did not show results. They refer to RNA-seq profiles in Fig 1a as a comment on the abundance of sense transcription in this cell line. If NET-CAGE data are clear only with the ED cell line, but not convincing with the TBX-4B cell line and primary samples ('data not shown') then the NET-CAGE section should be removed.

Reply:

As we described in the revised manuscript, we would like to propose the following two effects of the enhancer on the proviral expression.

(1) the enhancer affects antisense transcription by functionally interacting with the HTLV-1 3'LTR promoter

(2) the enhancer also affects both sense and antisense transcription by changing the chromatin openness of HTLV-1 provirus.

These should be the reason why there is a difference between wt-NFR and mut-NFR in some assay but not in other assays.

This is a speculation rather than a conclusion supported by strong experimental evidence. See Figs 4

Reply:

Expression level of each gene was analyzed by RNA-seq. The value is generated as TPM (transcript per million reads). To calculate TPM, we normalized for gene length first, and then normalize for sequencing depth second. Therefore, the upregulation observed is not due to overall expression.

I agree this is the usual - and widely accepted - method for quantifying expression levels generated by RNA-seq. My question is about the identity of these genes. Is this enhanced expression level affecting the vast majority of them or is there a bias towards a subset of genes, more specifically those located in the vicinity of integrated proviruses? I'm afraid I did not catch the authors' reply

Reply:

Thank you for critical and constructive comments. We do agree with the reviewer's point. Proviral and host genome expression is controlled not only by proviral sequence but also by integrated host genome. It is not correct to compare wt and mut in different clones, because their integration sites are different. Thus, we performed Crispr/Cas9 experiment to compare wt and mut HTLV-1 in same integration site. To confirm reproducibility, we generated additional clones having provirus in the same integration site as original clone but the NFR contains mutation. The result demonstrated that proviral expression was remarkably suppressed in the mutant clone. We showed the result as new data (Supplemental Fig. 5).

This is a convincing experiment. How about showing these data in the main results? It also complements the data in Fig 6 g, h.

Single cell data

Figs 6 g, h: I agree this shows transcription in the IS vicinity. Having added the cell numbers in the figures is also useful.

However, is this a direct effect of the enhancer on host gene transcription? The enhancer mutation studies demonstrate a decrease in provirus expression, HBZ, and as a result, the virus-host transcripts generated in the vicinity of the IS are suppressed given they depend on the production of HBZ. Is the effect of the mutation a direct effect on HBZ and is aberrant host transcription thus an indirect consequence? Is the conclusion (also in the title) "induces aberrant host gene expression" not an over-interpretation?

About HBZ expression in single cell data:

Reply:

We analyzed ATL8 data from the reviewer's point of view. Since single data is sparse, it is not easy to discuss about this point.

I perfectly understand and agree that scRNA-seq captures only a small proportion of the transcripts and that data analysis requires a critical number of cells to be processed. However, regarding HBZ in scRNA-seq data more specifically, the authors should be able to tell us in how many cells they find HBZ corresponding reads or patterns, even if it is none. Could they answer this question rather than send the reader to a bioRxiv paper?

Other comments:

Lines 88 ->

The results revealed an internal HTLV-1 enhancer, which has not been identified for 40 years since Poesz et al identified HTLV-1 in 1980 2.

This sentence should be removed. This is true for every new element discovered in HTLV-1 over many years of research since 1980 and new features are continuously revealed thanks to new technologies and the development of new methods. Examples are HBZ, CTCF ... and there will be more examples in the future. This is true for many other viruses and diseases.

Lines 343 ->

These findings indicate that the HTLV-1 enhancer can induce a 345 distinct alteration of the host transcriptome via chromatin looping, and thereby upregulates 346 cancer-related genes near ISs which might contribute to the preferential selection of a specific infected cell for clonal expansion during the early phase of leukemogenesis.

The statement that the enhancer is acting via chromatin looping is not supported by the authors' data. They show transcriptional changes at the edges of the provirus (at the virus-host junction or IS). They claim this is related to the presence of the enhancer. This is correct (indirect via HBZ production?). However, these transcriptional patterns are not a hallmark of chromatin looping. This sentence in the conclusion should be avoided as there is no experimental evidence for this.

Minor comments:

Several sentences in which subject-verb agreement is not correct. Example line 67: ATL cells exhibits Check the manuscript.

Lines 284 -> : "... but not in samples without ISs ... "These samples have ISs but they are not the specific IS corresponding to that of the transformed clone. Rephrasing this would clarify.

Lines 559-560: please rephrase or clarify

Reviewer #2 (Remarks to the Author):

In my opinion, the revised version of the manuscript has been greatly improved and the authors answered most of my concerns.

Minor comments:

- Line 42, typo : "The transcription factors, SRF and ELK-1, play a pivotal role..." (play and not plays)
- Figure 2c : the x-axis numbering is missing and the schemes are not aligned on the ticks of the y-axis.
- Figures 2C, suppl Fig 1 : please indicate rather $p < 0.001$ (and not $P = 0.000$).
- Please introduce the abbreviation for "integration site (IS)*" at line 151 rather than at line 214, i.e. the first time it is being mentioned.
- Please comment more on figure 3e.
- Line 226 : the sentence might be confusing as it may imply that each individual infected Jurkat cell contain hundreds of integration sites. It must be specified that you are talking about the cell condition infection, i.e. wt or mut_3. For example : "... in each condition, i.e. JET cells infected with HTLV-1-wt or mut_3 (Fig. 4h)".
- Line 232 : please capitalize the P of p-value.
- Supplementary figure 6 : please indicate for each pie, the number of total cells investigated.
- Line 315, typo : "... viral genes which play ... " (play and not plays).
- Line 407 : please, either use the plural or the line suffix, thus "JET cells are Jurkat cells expressing ... " or "The JET cell line consists of Jurkat cells expressing...". Also, please indicate a reference paper or company for this JET cell line. (you may visit the Cellosaurus website for help <https://www.ebi.ac.uk/ebiview/cellosaurus/>). This could be helpful as well for providing a reference to ED and TBX-4B cell lines.
- Line 437 : please replace "cultivation" by "culture"
- Lines 459, 476 and 493: "... using P5 and P7 primers..."
- Lines 488-491: please provide information on the amount or dilution of antibodies used.
- Lines 501-502 : please describe Jurkat cell transfection conditions (how many cells, how much DNA, how much transfection reagent).
- Line 511 : please specify how much biotin was added for the labeling.
- Line 518 : please specify how much DNA and how much transfection reagent.
- Line 520 : which volume of lysis buffer was used. Did you perform nuclear extraction right away or did you incubate briefly ? How much nuclear extraction buffer did you use ?
- Line 523 : please provide more detailed information about EMSA conditions, i.e. how much nuclear extract was used ? what is the molar concentration of probes used for competition?
- Line 534 ; please provide more details on the method : how much DNA, how much HilyMax ?
- Line 545 : gating strategy is presented in Suppl. Figure 8. Please write a legend that describes the strategy and describes the plots shown.
- Line 586 : if possible, please provide a reference justifying the choice of markers used.
- Line 596 : please specify the volume of RIPA used.
- Line 599 : please specify the volume of cell lysate used or protein equivalent. Lines 601-602 : specify the antibody concentration or dilution used for western blot.

Reviewer #3 (Remarks to the Author):

All concerns and suggestions were adequately addressed and the paper and figures were modified appropriately.

Matsuo et al: reply to reviewers' comments:

REVIEWER COMMENTS

Reviewer #1 (Remarks to the Author):

Overall, the authors have substantially improved the manuscript and the extra experiments strengthen the conclusions. However, the changes in the revised manuscript are not highlighted, as a result, it was complicated to navigate between the rebuttal and the revised manuscript and catch all changes mentioned by the authors. While the work has been significantly improved, it still comes with some over-interpretation and concerns described below.

Reply:

We appreciate the comment “the authors have substantially improved the manuscript and the extra experiments strengthen the conclusions”. We also appreciate the reviewer’s constructive comments and suggestions.

Main comments

Title:

“integrated genome”

The viral genome is integrated in the host genome, the host genome carries an integrated provirus, but the host genome is not “integrated”

I would suggest using a title close to the suggestion of reviewer 2, but leaving out “by a viral DNA seq capture approach” for the reasons explained in the first round of reviews.

Does the enhancer have a direct effect on host transcription or is this via viral gene expression (see comments below)?

Reply:

We have modified the title as the reviewer suggested. Now the title is “Identification and characterization of a novel enhancer in the HTLV-1 proviral genome”

Comments regarding some of the answers to our questions/comments:

Chromatin immunoprecipitation sequencing (CHIP-seq) signals of enhancer-related histone modifications, including H3K27Ac, H3K4me1, and H3K4me2, were high around the NFR in ED cells (Figure 2e, upper panel).

What is meant by around?

1st Reply:

We corrected this from “around” to “in the proximity of” in the revised manuscript

Can the authors provide a more quantitative explanation? How distant is “proximity”?

2nd Reply:

We provided distance in the revised manuscript as below.

Chromatin immunoprecipitation sequencing (ChIP-seq) signals of enhancer-related histone modifications, including H3K27Ac, H3K4me1, and H3K4me2, showed peaks within a 3-kb distance from the NFR in ED cells (Fig. 2e and f), whereas the peaks were not observed in infected clone without HTLV-1 integration in this locus (Supplementary Fig. 2a).

NET-CAGE data

In their reply, the authors mention they performed NET-CAGE on the TBX-4B cells, but did not show results. They refer to RNA-seq profiles in Fig 1a as a comment on the abundance of sense transcription in this cell line. If NET-CAGE data are clear only with the ED cell line, but not convincing with the TBX-4B cell line and primary samples (‘data not shown’) then the NET-CAGE section should be removed.

Reply:

We showed the data in Supplementary Figure 2d.

Reply:

As we described in the revised manuscript, we would like to propose the following two effects of the enhancer on the proviral expression.

(1) the enhancer affects antisense transcription by functionally interacting with the HTLV-1 3’LTR promoter

(2) the enhancer also affects both sense and antisense transcription by changing the chromatin openness of HTLV-1 provirus.

These should be the reason why there is a difference between wt-NFR and mut-NFR in some assay but not in other assays.

This is a speculation rather than a conclusion supported by strong experimental evidence. See Figs 4

Reply:

We do understand the reviewer's point. We added the sentence "Further experiments are required to elucidate how these viral regulatory elements, including 5'LTR, viral insulator, viral enhancer and 3'LTR, in the small HTLV-1 provirus cooperatively regulate viral and host genome transcriptome."

Reply:

Expression level of each gene was analyzed by RNA-seq. The value is generated as TPM (transcript per million reads). To calculate TPM, we normalized for gene length first, and then normalize for sequencing depth second. Therefore, the upregulation observed is not due to overall expression.

I agree this is the usual - and widely accepted - method for quantifying expression levels generated by RNA-seq. My question is about the identity of these genes. Is this enhanced expression level affecting the vast majority of them or is there a bias towards a subset of genes, more specifically those located in the vicinity of integrated proviruses? I'm afraid I did not catch the authors' reply

Reply:

We performed PCA analysis to investigate global transcriptional differences among JET clones used in the analysis. The result suggested that there was much less global transcriptional difference among various JET clones when compared with TBX-4B and ED cells (new Fig. 5d). We have also compared the expression level of viral genes, *tax* and *HBZ*, among each clone (new Supplementary Fig. 5a). Further, we analyzed publicly available RNA-seq data regarding Jurkat cells containing inducible *tax* or *HBZ*

gene to investigate whether the genes we show in Fig. 5e would be induced by *tax* or *HBZ* expression. The result indicated that there is little bias affected by viral gene expression in the upregulated genes in Fig. 5e. We show this data as new Supplementary Fig. 5b. These data indicated that the result we showed in figure 5d is likely due to the effect *in cis* caused by HTLV-1 integration. However, we cannot exclude other possibilities completely. Thus, we have added the sentence below. These data indicated that up-regulation of the host genes near viral IS was not via viral gene expression but at least partially mediated by ectopic present of the enhancer inserted by HTLV-1. Further experiments are needed to understand the effect of HTLV-1 integration on the host genome.

Reply:

Thank you for critical and constructive comments. We do agree with the reviewer's point. Proviral and host genome expression is controlled not only by proviral sequence but also by integrated host genome. It is not correct to compare wt and mut in different clones, because their integration sites are different. Thus, we performed Crispr/Cas9 experiment to compare wt and mut HTLV-1 in same integration site. To confirm reproducibility, we generated additional clones having provirus in the same integration site as original clone but the NFR contains mutation. The result demonstrated that proviral expression was remarkably suppressed in the mutant clone. We showed the result as new data (Supplemental Fig. 5).

This is a convincing experiment. How about showing these data in the main results? It also complements the data in Fig 6 g, h.

Reply:

Thank you for valuable suggestions. I do agree the reviewer's point. We have re-organized figures according to the reviewer's comment. We think that figures in revised manuscript become clearer and more convincing than previous version.

Single cell data

Figs 6 g, h: I agree this shows transcription in the IS vicinity. Having added the cell numbers in the figures is also useful.

However, is this a direct effect of the enhancer on host gene transcription? The enhancer mutation studies demonstrate a decrease in provirus expression, HBZ, and as a result, the virus-host transcripts generated in the vicinity of the IS are suppressed given they depend on the production of HBZ. Is the effect of the mutation a direct effect on HBZ and is aberrant host transcription thus an indirect consequence? Is the conclusion (also in the title) “induces aberrant host gene expression” not an over-interpretation?

Reply:

We really appreciate the reviewer’s careful interpretation and thoughtful comments. We also think this point again carefully. We have added cell numbers in the revised figure. Regarding Figs 7g, h, aberrant transcriptions were observed the host genome next to HTLV-1 IS, indicating the effect is likely to be induced by inserted provirus in cis not via viral genes in trans.

In addition, we have changed the title to avoid the criticism that there is an over-interpretation.

About HBZ expression in single cell data:

Reply:

We analyzed ATL8 data from the reviewer’s point of view. Since single data is sparse, it is not easy to discuss about this point.

I perfectly understand and agree that scRNA-seq captures only a small proportion of the transcripts and that data analysis requires a critical number of cells to be processed. However, regarding HBZ in scRNA-seq data more specifically, the authors should be able to tell us in how many cells they find HBZ corresponding reads or patterns, even if it is none. Could they answer this question rather than send the reader to a bioRxiv paper?

Reply:

We analyzed how many cells express HBZ in ATL1 and ATL8. We found that 107 cells and 4,598 cells were positive for HBZ in ATL1 and ATL8, respectively. Now our single

cell study has been published in The Journal of Clinical Investigation. We added the paper as a reference paper in the revised manuscript for the readers.

Other comments:

Lines 88 ->

The results revealed an internal HTLV-1 enhancer, which has not been identified for 40 years since Poiesz et al identified HTLV-1 in 1980 2.

This sentence should be removed. This is true for every new element discovered in HTLV-1 over many years of research since 1980 and new features are continuously revealed thanks to new technologies and the development of new methods. Examples are HBZ, CTCF ... and there will be more examples in the future. This is true for many other viruses and diseases.

Reply:

What we would like to emphasize here is that there are many studies have been performed regarding HTLV-1 tax gene. There are 2,213 results found when we search "htlv and tax" by Pubmed. Since the enhancer is located in the coding region of tax gene, the researcher used tax expressing construct without knowing there is the enhancer region in the DNA. The sentence would be helpful for the readers who are not so familiar with the virus to get the point.

However, we are all ears for the editor and the reviewer's opinion. If the reviewer think we still should remove this sentence, we are going to do that.

Lines 343 ->

These findings indicate that the HTLV-1 enhancer can induce a
345 distinct alteration of the host transcriptome via chromatin looping, and thereby upregulates
346 cancer-related genes near ISs which might contribute to the preferential selection of a specific infected cell for clonal expansion during the early phase of leukemogenesis.

The statement that the enhancer is acting via chromatin looping is not supported by the authors' data. They show transcriptional changes at the edges of the provirus (at the virus-host junction or IS). They claim this is related to the presence of the enhancer. This is correct (indirect via HBZ production?). However, these transcriptional patterns are not a hallmark of chromatin looping. This sentence in the conclusion should be avoided as there is no experimental evidence for this.

Reply:

We appreciated for the reviewer's valuable suggestion. To avoid the misleading that we show chromatin looping via CTCF, we clearly described which study demonstrated that as below.

HTLV-1 contains CTCF-binding sites and therefore viral integration generates an ectopic CTCF-binding site in the host genome¹¹, which induces deregulation of host gene transcription via chromatin looping as Melamed A et al reported²⁶.

Minor comments:

Several sentences in which subject-verb agreement is not correct. Example line 67: ATL cells exhibits Check the manuscript.

Lines 284 -> : "... but not in samples without ISs ... "These samples have ISs but they are not the specific IS corresponding to that of the transformed clone. Rephrasing this would clarify.

Reply:

We do apologize for these mistakes. We have corrected them in the revised manuscript.

Lines 559-560: please rephrase or clarify

Reply:

We have corrected them in the revised manuscript.

Reviewer #2 (Remarks to the Author):

In my opinion, the revised version of the manuscript has been greatly improved and the authors answered most of my concerns.

Reply: We appreciate the comment “the revised version of the manuscript has been greatly improved”.

Minor comments:

- Line 42, typo : “The transcription factors, SRF and ELK-1, play a pivotal role...” (play and not plays)

Reply: We corrected this point.

- Figure 2c : the x-axis numbering is missing and the schemes are not aligned on the ticks of the y-axis.

Reply: We corrected this point.

- Figures 2C, suppl Fig 1 : please indicate rather $p < 0.001$ (and not $P = 0.000$).

Reply: We corrected this point.

- Please introduce the abbreviation for “integration site (IS)*” at line 151 rather than at line 214, i.e. the first time it is being mentioned.

Reply: We corrected this point.

- Please comment more on figure 3e.

Reply: We have added the following text.

The result showed that there was clear competition observed with unlabeled wt probes but the competition activity was remarkably decreased when we used mut_1, mut_2 and mut_3 probes.

- Line 226 : the sentence might be confusing as it may imply that each individual

infected Jurkat cell contain hundreds of integration sites. It must be specified that you are talking about the cell condition infection, i.e. wt or mut_3. For example : "... in each condition, i.e. JET cells infected with HTLV-1-wt or mut_3 (Fig. 4h)".

Reply: We corrected this point according to the reviewer's suggestion.

- Line 232 : please capitalize the P of p-value.

Reply: We corrected this point.

- Supplementary figure 6 : please indicate for each pie, the number of total cells investigated.

Reply: We corrected this point.

- Line 315, typo : "... viral genes which play ... " (play and not plays).

Reply: We corrected this point.

- Line 407 : please, either use the plural or the line suffix, thus "JET cells are Jurkat cells expressing ... " or "The JET cell line consists of Jurkat cells expressing...". Also, please indicate a reference paper or company for this JET cell line. (you may visit the Cellosaurus website for help <https://www.expasy.org/resources/cellosaurus>). This could be helpful as well for providing a reference to ED and TBX-4B cell lines.

Reply: We corrected these points.

- Line 437 : please replace "cultivation" by "culture"

Reply: We corrected this point.

- Lines 459, 476 and 493: "... using P5 and P7 primers..."

Reply: We corrected these points.

- Lines 488-491: please provide information on the amount or dilution of antibodies used.

Reply: We corrected this point.

- Lines 501-502 : please describe Jurkat cell transfection conditions (how many cells, how much DNA, how much transfection reagent).

Reply: We corrected this point.

- Line 511 : please specify how much biotin was added for the labeling.

- Line 518 : please specify how much DNA and how much transfection reagent.

Reply: We corrected this point.

- Line 520 : which volume of lysis buffer was used. Did you perform nuclear extraction right away or did you incubate briefly ? How much nuclear extraction buffer did you use ?

Reply: We corrected this point.

- Line 523 : please provide more detailed information about EMSA conditions, i.e. how much nuclear extract was used ? what is the molar concentration of probes used for competition?

Reply: We corrected this point.

- Line 534 ; please provide more details on the method : how much DNA, how much HilyMax ?

Reply: We corrected this point.

- Line 545 : gating strategy is presented in Suppl. Figure 8. Please write a legend that describes the strategy and describes the plots shown.

Reply: We corrected this point.

- Line 586 : if possible, please provide a reference justifying the choice of markers used.

Reply: We corrected this point.

- Line 596 : please specify the volume of RIPA used.

Reply: We corrected this point.

- Line 599 : please specify the volume of cell lysate used or protein equivalent. Lines 601-602 : specify the antibody concentration or dilution used for western blot.

Reply: We corrected these points.

Reviewer #3 (Remarks to the Author):

All concerns and suggestions were adequately addressed and the paper and figures were modified appropriately.

Reply: We really appreciated the reviewer #3 for taking precious time to review this manuscript.

REVIEWER COMMENTS, third round

Reviewer #1 (Remarks to the Author):

I am satisfied with the authors' answers to my comments and appreciate the modifications made in the revised manuscript. The additional figures or their modified organization further address the issues raised and nicely clarify the remaining questions.

I would like to congratulate the authors. This is a very nice piece of work. Also, I believe that having brought in additional work and new data according to the reviewers' comments was worth the effort. It has paid off and resulted in a high quality paper.

The only point I still do not agree with is the sentence discussed in the rebuttal:
"The results revealed an internal HTLV-1 enhancer, which has not been identified for 40 years since Poiesz et al identified HTLV-1 in 1980."

Although I perfectly get the authors' reply (copied below), this is true for many other elements that have been discovered in HTLV-1 over the previous years. Among other examples, studies using cloned proviruses before the discovery of HBZ illustrate this point. Conclusions were drawn from such studies without knowing HBZ had a substantial impact on these observations. The authors could replace this sentence by a statement pointing to the impact this enhancer may have had in previous studies (example cited by the authors: tax-expressing constructs) without being recognized at that time.

(Reviewer's comment and authors' reply):

This sentence should be removed. This is true for every new element discovered in HTLV-1 over many years of research since 1980 and new features are continuously revealed thanks to new technologies and the development of new methods. Examples are HBZ, CTCF ... and there will be more examples in the future. This is true for many other viruses and diseases.

Reply:

What we would like to emphasize here is that there are many studies have been performed regarding HTLV-1 tax gene. There are 2,213 results found when we search "htlv and tax" by Pubmed. Since the enhancer is located in the coding region of tax gene, the researcher used tax expressing construct without knowing here is the enhancer region in the DNA. The sentence would be helpful for the readers who are not so familiar with the virus to get the point. However, we are all ears for the editor and the reviewer's opinion. If the reviewer think we still should remove this sentence, we are going to do that

Reviewer #2 (Remarks to the Author):

All my concerns have been addressed satisfactorily, as well as the ones from the other reviewers, and the resulting manuscript has been greatly improved.

Matsuo et al: reply to reviewers' comments:

REVIEWER COMMENTS

Reviewer #1 (Remarks to the Author):

I am satisfied with the authors' answers to my comments and appreciate the modifications made in the revised manuscript. The additional figures or their modified organization further address the issues raised and nicely clarify the remaining questions.

I would like to congratulate the authors. This is a very nice piece of work. Also, I believe that having brought in additional work and new data according to the reviewers' comments was worth the effort. It has paid off and resulted in a high quality paper.

Reply: We really appreciated the reviewer #1 for taking precious time to review this manuscript.

The only point I still do not agree with is the sentence discussed in the rebuttal: "The results revealed an internal HTLV-1 enhancer, which has not been identified for 40 years since Poiesz et al identified HTLV-1 in 1980."

Although I perfectly get the authors' reply (copied below), this is true for many other elements that have been discovered in HTLV-1 over the previous years. Among other examples, studies using cloned proviruses before the discovery of HBZ illustrate this point. Conclusions were drawn from such studies without knowing HBZ had a substantial impact on these observations. The authors could replace this sentence by a statement pointing to the impact this enhancer may have had in previous studies (example cited by the authors: tax-expressing constructs) without being recognized at that time.

(Reviewer's comment and authors' reply):

This sentence should be removed. This is true for every new element discovered in

HTLV-1 over many years of research since 1980 and new features are continuously revealed thanks to new technologies and the development of new methods. Examples are HBZ, CTCF ... and there will be more examples in the future. This is true for many other viruses and diseases.

Reply:

What we would like to emphasize here is that there are many studies have been performed regarding HTLV-1 tax gene. There are 2,213 results found when we search "htlv and tax" by Pubmed. Since the enhancer is located in the coding region of tax gene, the researcher used tax expressing construct without knowing here is the enhancer region in the DNA. The sentence would be helpful for the readers who are not so familiar with the virus to get the point. However, we are all ears for the editor and the reviewer's opinion. If the reviewer think we still should remove this sentence, we are going to do that.

Reply: We have changed the sentence in the revised manuscript as below.

The results revealed an internal viral enhancer in the HTLV-1 provirus. It is noteworthy that the region has been intensively analyzed as a coding region of the oncogenic viral gene *tax* without knowing the enhancer function.

Reviewer #2 (Remarks to the Author):

All my concerns have been addressed satisfactorily, as well as the ones from the other reviewers, and the resulting manuscript has been greatly improved.

Reply: We really appreciated the reviewer #2 for taking precious time to review this manuscript.